Registered report

psychology

prediction markets, peer beliefs, hypothesis

**Author for correspondence:**
Anna Dreber
e-mail: anna.dreber@hhs.se

# Using prediction markets to predict the outcomes in the Defense Advanced Research Projects Agency's next-generation social science programme

Domenico Viganola[1], Grant Buckles[2], Yiling Chen[3], Pablo Diego-Rosell[2], Magnus Johannesson[4], Brian A. Nosek[5,6], Thomas Pfeiffer[7], Adam Siegel[8] and Anna Dreber[4,9]

[1]World Bank Group, Washington D.C., USA
[2]Gallup Inc, Washington, District of Columbia, USA
[3]Paulson School of Engineering and Applied Sciences, Harvard University, Cambridge, MA, USA
[4]Department of Economics, Stockholm School of Economics, Stockholm, Sweden
[5]Center for Open Science, Charlottesville, VA, USA
[6]Department of Psychology, University of Virginia, Charlottesville, VA, USA
[7]New Zealand Institute for Advanced Study, Massey University, Auckland, New Zealand
[8]Cultivate Labs, Chicago, IL, USA
[9]Department of Economics, University of Innsbruck, Innsbruck, Austria

GB, 0000-0001-5901-4317; TP, 0000-0002-0592-577X;
AD, 0000-0003-3989-9941

There is evidence that prediction markets are useful tools to aggregate information on researchers' beliefs about scientific results including the outcome of replications. In this study, we use prediction markets to forecast the results of novel experimental designs that test established theories. We set up prediction markets for hypotheses tested in the Defense Advanced Research Projects Agency's (DARPA) Next Generation Social Science (NGS2) programme. Researchers were invited to bet on whether 22 hypotheses would be supported or not. We define support as a test result in the same direction as hypothesized, with a Bayes factor of at least 10 (i.e. a likelihood of the observed data being consistent with the tested hypothesis that is at least 10 times greater compared with the null hypothesis). In addition to betting on this binary outcome, we asked participants to bet

on the expected effect size (in Cohen's $d$) for each hypothesis. Our goal was to recruit at least 50 participants that signed up to participate in these markets. While this was the case, only 39 participants ended up actually trading. Participants also completed a survey on both the binary result and the effect size. We find that neither prediction markets nor surveys performed well in predicting outcomes for NGS2.

## 1. Introduction

Prediction markets can be used to aggregate private information [1]. By trading contracts with payoffs that depend on clearly defined outcomes, market participants can generate forecasts about future events. The price of a contract with a binary event can, with some caveats [2] be interpreted as the probability the market assigns to this event. Prediction markets have successfully predicted outcomes in sports, entertainment and politics [3–6]. Markets on scientific studies were first suggested by Robin Hanson [7] in the paper 'Could gambling save science? Encouraging an honest consensus', and later tested in the laboratory by Almenberg *et al.* [8]. Accumulating evidence suggests that prediction markets are effective for accurately identifying which results will replicate in replication studies [9–12]. As reported in Gordon *et al.* [13], pooling the results from four prediction market studies on binary outcomes (whether the study replicated or not; table 1) gives a 73% (76/104) correct prediction rate if we interpret prices above 50 as market prediction for successful replication. The corresponding number for a survey measure similar to the one we use here is 66% (68/103). The peer beliefs about replicability estimated in prediction markets can thus be viewed as a reproducibility indicator, and allows us to assess additional information such as prior and posterior probabilities of the hypotheses being tested.

So far, prediction market studies in science focused on large-scale systematic replication projects. An advantage of these projects is that they provide a large set of studies that use similar methodologies, are conducted within a near future, and are documented in detail prior of being conducted. With the emergence of large-scale, crowd-sourced projects beyond direct replication [15], there is now the opportunity to explore prediction markets in science in a broader context.

One such opportunity is provided by the Defense Advanced Research Projects Agency's (DARPA) Next Generation Social Science (NGS2) programme. NGS aims to develop new methods, tools and techniques to address research questions concerning complex social systems, which are difficult or impossible to answer with current methods. When examined individually, each theory arising from research on social systems may seem straightforward and plausible. When reviewed collectively, however, it becomes apparent that theories are mutually incoherent and models are logically incompatible, giving rise to the so-called 'incoherency problem' [16]. Current methods in social science research fail to effectively disentangle these incoherencies and to answer the question of 'what matters most?' for the emergence of social phenomena in different contexts.

NGS2 research groups developed the technical capabilities to perform experimental research at the scale necessary, and with sufficient control, to discern between the multiple interacting factors that impact social phenomena. Simultaneously, the NGS2 programme addressed the intended practical application of these efforts from a different angle. When decision makers in government agencies, private companies and academic institutions turn to the myriad literature on complex social systems to inform new policies or other decisions, they confront the incoherency among theories and are no closer to making well-informed decisions. This problem may be addressed by decreasing the uncertainty in theories using a collaborative, crowd-sourced method to improve reasoning on the central hypotheses for a given context.

In this study, we use prediction markets and survey measures to estimate peer beliefs about the outcomes from the NGS2 research group. In Cycle 3 of the programme, Diego-Rosell *et al.* studied group innovation in competitive or uncertain environments, with a design that tested different theories against one another [17]. Diego-Rosell *et al.* planned to recruit up to 5000 participants to a multi-player online gaming platform where participants will face a resource maximization challenge.[1]

---

[1]Participants in the NGS2 experiments were recruited from contacts collected through the Gallup Panel, Gallup World Poll, Gallup marketing and Volunteer Science. Invitations were sent via email and mobile notifications (for those who opted in). Volunteer Science also directed advertisements via Facebook, Twitter, Google and Mechanical Turk. Participants were provided with links to

**Table 1.** Prediction market studies to forecast the outcomes of systematic replication studies. Traders were endowed with US$100 [9] or US$50 [10–12], and bet on the binary outcome of whether a study would replicate or not by trading contracts worth $1 in case of successful replication, and $0 otherwise. (Successful replication was mainly defined as an effect in the same direction as the original study with a *p*-value less than 0.05.) ML2 additionally elicited predictions on the effect sizes in the replications, and SSRP had two treatments to adjust to the design used for the replications. Trading periods ranged from 10 days to two weeks, and traders were given detailed information about the original studies and the replications.

| | | number of replications in market | participants in the market | references | key characteristics |
|---|---|---|---|---|---|
| reproducibility project: psychology | RPP | 23 (Period 1) | 47 (Period 1) | replication results: [14] | Studies were selected from top psychology journals in 2008. Prediction markets were conducted in two periods on a subset of all studies replicated in RPP. Traders were recruited through the email lists of the Open Science Framework (OSF) and the RPP collaboration. |
| | | 21 (Period 2) (results for 41 out of 44 ready and analysed) | 45 (Period 2) | market: [9] | |
| experimental economics replication project | EERP | 18 | 97 | [10] | Experimental economics studies were selected from top economics journals 2011–2014. Traders were recruited mainly through the Economic Science Association email list. |
| social science replication project | SSRP | 21 | 114 (Treatment 1) 92 (Treatment 2) | [12] | Experimental social science studies were selected from *Nature* and *Science* 2010–1025. Multiple social science mailing lists and tweets were used for recruitment. |
| ManyLabs2 | ML2 | 28 (results for 24 analysed) | 78 | [11] | Twenty-eight new and classic effects in psychology were selected. Traders were recruited mainly through the email list of the Open Science Framework (OSF). |

Diego-Rosell *et al*. used a between-participant experimental design with repeated measures, in which the outcome variable is the group motivation to innovate, on two platforms: Polycraft World and Boomtown. We set out to not predict all of these experiments but a subset of 22 hypotheses tested on Boomtown; for these 22 hypotheses, we set up prediction markets and survey measures.

We invited researchers to bet on whether the 22 hypotheses will be supported or not, with 'supported' being defined as a test result in the same direction as the hypothesis, with a Bayes factor of at least 10. In addition to betting on this binary outcome for each hypothesis, we asked researchers to bet on the standardized effect size in Cohen's $d$ units for each hypothesis. Participants also filled out a survey on the binary results and effect sizes before the prediction markets started. This project is part of a cumulative effort across large samples of pre-registered studies to collect prediction data on scientific outcomes (e.g. [9–12]). We are accumulating data across studies to together get a sizeable sample size for evaluating the general applicability of prediction markets and surveys and examine factors associated with their predictive validity. A unique contribution of this study in this programme of research is that we go beyond forecasting direct replications and focus on the applicability and generalizability of theories that are tested with novel experimental designs.

# 2. Methods

## 2.1. Participants

Participants were recruited through social media via Brian Nosek's twitter feed and other behavioural science organizations. Brian Nosek has a large following of researchers that are interested in reproducibility, and this method has been successful in related previous studies at attracting a sizeable sample of participants. We expected participants to be mainly researchers in psychology and the behavioural sciences and interested in open science and reproducibility, and we required them to have at least an MSc degree or to be working on related topics. We set out to recruit at least 50 participants that make at least one trade on the prediction markets—if less people signed up our plan was to not run the markets (as described in the invitation letter). While we did indeed get more than 50 participants signing up so that we could run the markets, fewer actually finished the survey and traded (see more below). We asked participants to electronically give their consent when sending them the pre-market survey. Only the participants that filled out the survey were entitled to participate in the markets. In our analysis we included only survey data of participants that did at least one trade in the markets. If one or several of the 22 hypotheses investigated in the NSG2 programme would have ended up not being tested, we would not have complete data for that specific hypothesis/hypotheses and would thus have adjusted the analysis accordingly by excluding this hypothesis from the analysis. This ended up not being the case.

## 2.2. Survey measures

Before participating in the prediction markets, participants answered survey questions where, for each hypothesis, they were asked to give the probability (on a scale from 0% to 100%) that the hypothesis will be supported by an effect in the hypothesized direction with a Bayes factor of at least 10 if compared with a pre-specified alternative hypothesis. Participants were asked 'What is the probability that the hypothesis H1 [description of H1] is supported by the data collected in the Boomtown experiment, when compared to hypothesis H0.1 [description of H0.1]? Hypothesis H1 is supported if according to the data, the likelihood of H1 is 10 times greater or higher than the likelihood of H0.1 under the following priors: [details on the priors, including effects and standard deviations, of H1 and H0.1]'. We also asked participants to predict the effect size in terms of Cohen's $d$. Participants were asked 'What will be the effect of [independent variable of interest] on the [dependent variable of interest], in terms of Cohen's $d$?' We reminded participants about which effect sizes are considered to be small, medium and large in terms of Cohen's $d$.

download the Boomtown app from the Apple iStore or Google Play Store as part of their invitation. For all sources, participants were screened to ensure eligibility. All participants had to be at least 18 years of age, have Internet access and speak English. Data collection was carried out in two cycles, with a first cycle conducted between February and March 2020, yielding a total of 438 completed experiments, and a second cycle conducted in August 2020 with 1118 completed experiments. The cumulative total of 1546 experiments included a total of 3293 participants and 127 085 valid individual decisions.

Both measures were incentivized with a quadratic scoring rule. The participants were told that one participant would be randomly picked and paid for the binary measure and one participant would be randomly picked and paid for the effect size forecast.

For the binary measure payment, the randomly picked participant would for a randomly picked hypothesis receive $100 \times (1 - (p^* - p_i)^2)$ where $p^*$ is 1 if the hypothesis is supported and 0 if not and $p_i$ is the probability that individual $i$ assigns to the hypothesis being supported. For the effect size payment, the randomly picked participant received $100 -$ (average of the mean squared error $\times 100$), where the average of the mean squared error is the average of the squared differences between the participant's answers and the actual effect sizes for all of the 22 hypotheses (in terms of Cohen's $d$).

In the survey, we also collected some demographic information about the participants. We asked them about whether they are active researchers in academia (response options: not in academia, bachelor student, master student, PhD student, post-doc or assistant professor, lecturer or associate professor, full professor) and their expertise in the general topic of the hypotheses tested in NGS2 ('Please rate your status of expertise for the hypotheses tested in NGS2 on a scale from 1 (no knowledge of the topic) to 7 (very high knowledge of the topic)').

## 2.3. Prediction markets

The study was conducted through a web-based prediction market interface that was open for two weeks. During this time period, participants could voluntarily at any point make decisions on whether to trade on a hypothesis. Before starting to trade, participants received information about the trading procedure and answered three comprehension questions about the functioning of the markets. We had two sets of markets on separate trading platforms—one for whether the test result supports the hypothesis or not (binary outcome) and one for the effect size. Participants were endowed with a total of $50 split into endowments of $25 on each trading platform (expressed as 10 000 points on each platform). These points could be used to make predictions.

Predictions were made by buying and selling stocks on the hypotheses on an interface that highlights the forecasting functionality of the market. When participants bet on whether the test result supports the hypothesis or not with a Bayes factor of at least 10, participants traded contracts that pay 100 points if the test result supports the hypothesis and 0 points otherwise.

For each hypothesis, participants could see the current market prediction for the probability of successful support for a hypothesis and the probability for various effect sizes. The trading platforms used an automated market maker implementing a logarithmic market scoring rule [18]. This algorithm offered a buying price and a selling price at all times and thus made sure that there is always a counterparty to trade with. The algorithm used the net sales ($s$) the market maker had done so far in a market to determine the prices for an infinitesimally small trade as $p = 100 \times \exp(s/b)/(\exp(s/b) + 1)$, where $b$ is the liquidity parameter that we set to 50. To buy (sell) shares in a hypothesis, participants chose how many shares to go long (short) with on the trading interface and could see the cost of this trade, or instead first chose how much money to invest and then saw the amount of shares this corresponded to. Each additional point invested in long (short) increased (decreased) the price. The market maker was set up so that the value of a long share is 100 minus the value of a short share. Participants could also increase or decrease existing positions. The starting price was 50.

For effect sizes the logic was similar but here participants were asked to predict effect sizes in terms of Cohen's $d$ in the range of $[-1, +1]$. We mapped this range to contracts with payouts between 0 and 100 to avoid negative contract prices. The contract paid to its holder $50 \times (X + 1)$, where $X$ is the observed effect size. Hence the lowest realized value of this contract was 0 (when $X = -1$) and the highest was 100 (when $X = 1$). Participants could do short trades, with short positions paying $50 \times (1 - X)$, and have the lowest realized value of 0 (when $X = 1$) and the highest realized value of 100 (when $X = -1$). We used the same market maker as above, which ensures that the prices of a short and a long always add up to 100. The starting price was 50, corresponding to an effect size $X$ of zero.

The prediction markets were settled in September 2020.

## 2.4. Specific hypotheses and analysis

Following recent recommendations [19], we consider results with $p$-values below 0.005 to be statistically significant in our hypotheses tests specified below and results with $p$-values below 0.05 as suggestive evidence. For the primary hypotheses, we use Spearman correlations which are based on ranks (assuming a monotonic, but not necessarily linear relationship, between beliefs and outcomes). We have

no reason to expect a non-monotonic relationship. All *p*-values are two-sided. If some participants participated only in the survey and did not make at least one trade on the prediction markets, they were not included in the hypothesis tests or exploratory analyses based on the survey responses below. This is to ensure that the prediction market data and the survey data are based on the same individuals.

## 2.5. Statistical power

While the reliability of survey answers and prediction market prices on the market level was likely to be high with what we thought would be around 50 traders and many trades, the sample size of 22 NGS2 studies is small and all results should thus be interpreted with caution. The limited number of NGS2 studies included ($n = 22$) leads to low statistical power in the hypotheses outlined below. But this study is part of an ongoing cumulative project collecting data on the performance of prediction markets and surveys to predict the results of scientific studies. We have used prediction markets in four previous studies to predict replication outcomes [9–12]. The correlation between prediction market beliefs and replication outcomes was 0.42 (RPP), 0.30 (EERP), 0.84 (SSRP) and 0.76 (ML2) in these four studies. If we carry out a power calculation for the average correlation of 0.58 of these four studies (as reported in [13]) on replication outcomes for the first and third primary hypotheses below, we have 51% power to detect a correlation of 0.58 at the 0.005 level for 'statistical significance' and 83% power at the 0.05 level for 'suggestive evidence'. In another project (the crowdsourcing a hypothesis test (CAHT) project [20]) we used an incentivized survey to predict effect sizes for new studies and we observed a correlation of 0.71 between predicted and observed effect sizes; this previous result is relevant as a meter of comparison for the second and fourth primary hypotheses below. If we carry out a power calculation for a correlation of 0.71, we have 84% power to detect an effect at the 'statistical significance' ($p = 0.005$) level and 97% power for 'suggestive evidence' ($p = 0.05$). The statistical power of finding support for our hypotheses below is thus medium and it is not clear to what extent it matters if the power calculations are based on previous work on replications or on previous work on new outcomes, but the data from this study will also be used in our cumulative effort to collect data on the ability of prediction markets and surveys to predict scientific results.

## 2.6. Primary hypothesis 1: prediction markets and binary outcomes

H0: Prediction market final prices are not better than random in predicting whether the test results support the hypotheses or not with *p*-values less than 0.005.
H1: Prediction market final prices are significantly better than random in predicting whether the test results support the hypotheses or not with *p*-values less than 0.005.

This is tested by a Spearman correlation, where we correlate prediction market prices with the binary outcome whether the hypotheses are supported or not. If there is a significant correlation in the correct direction, we interpret this as evidence against H0.

## 2.7. Primary hypothesis 2: prediction markets and effect sizes

H0: Prediction market final prices are not better than random in predicting effect sizes.
H1: Prediction market final prices are significantly better than random in predicting effect sizes.

This is tested by a Spearman correlation, where we correlate predicted effect sizes from the prediction market with actual effect sizes. If there is a significant correlation in the correct direction, we interpret this as evidence against H0.

## 2.8. Primary hypothesis 3: survey answers and binary outcomes

H0: Survey predictions are not better than random in predicting whether the test results support the hypotheses or not with *p*-values less than 0.005.
H1: Survey predictions are significantly better than random in predicting whether the test results support the hypotheses or not with *p*-values less than 0.005.

This is tested by a Spearman correlation, where we correlate survey probabilities with the binary outcome whether the hypotheses are supported or not. If there is a significant correlation in the correct direction, we interpret this as evidence against H0. As the prediction market data by definition are aggregated for each of the 22 hypotheses investigated in the NSG2 programme (one hypothesis is one market), we will average the survey data over all participants for both primary hypotheses 3 and 4.

## 2.9. Primary hypothesis 4: survey answers and effect sizes

H0: Survey predictions are not better than random in predicting effect sizes.
H1: Survey predictions are significantly better than random in predicting effect sizes.

This is tested by a Spearman correlation, where we correlate predicted effect sizes from the survey with actual effect sizes. If there is a significant correlation in the correct direction, we interpret this as evidence against H0.

## 2.10. Secondary hypotheses

We compare the absolute prediction error between the prediction market and the average survey for the 22 hypotheses, for both the binary measure and the effect size measure, using a Wilcoxon signed-ranks test. As a robustness check, we perform the same comparison using the squared prediction error (Brier score) as a measure of prediction accuracy.

Finally, we also test if the average predictions differ from the outcomes (where our hypothesis is the null hypothesis of no difference). This is done with a Wilcoxon signed-ranks test. We compare the prediction market predictions on the binary outcomes with the actual outcomes; the prediction market predictions on effect sizes compared with actual effect sizes; survey predictions on the binary outcomes with the actual outcomes; and survey predictions on effect sizes compared with actual effect sizes.

## 2.11. Exploratory analyses

We test if two demographic variables (academia versus non-academia and expertise in the hypotheses) are associated with survey predictions. We test if these two variables are statistically significantly associated with the individual absolute prediction error on survey predictions, in an ordinary least-squares individual data regression with clustering of standard errors on the individuals to take into account the correlation of multiple survey responses within individuals. The background variable about academia is included as a dummy variable with 1 = *in academia* (including students) and 0 = *not in academia* and the question about expertise in the topic tested in the hypothesis is included as a continuous variable with the answer on the question 'Please rate your status of expertise for the hypotheses tested in NGS2 on a scale from 1 (no knowledge of the topic) to 7 (very high knowledge of the topic)' used to assess expertise.

## 2.12. Robustness analyses

Findings from similar prediction markets indicate that most of the information gets priced in soon after the markets opened. Prices then tend to fluctuate around equilibrium prices. To minimize the impact of noisy price fluctuations on our forecasts, we use a time-weighted average of the prices from the second half of the trading period as alternative market forecasts. We run the tests specified above using these smoothed market prices instead of the final market prices.

# 3. Results

Thirty-nine (39) participants completed the survey and did at least one trade in either the binary outcome market or the effect size market. This includes 3 full professors, 6 lecturer or associate professors, 17 post-doctoral researchers or assistant professors, 8 PhD students and 5 participants selecting 'Not in academia'.

For the binary outcomes, the final market forecasts ranged from 0.38 to 0.89, with a mean of 0.58, and the survey-based forecasts ranged from 0.36 to 0.76, with a mean of 0.54. For the effect sizes, the final market forecasts ranged from −0.18 to 0.8, with a mean of 0.17, and the survey-based forecasts ranged from −0.01 to 0.44, with a mean of 0.16 (table 2).

There is no statistically significant correlation between final market forecast and observed outcome for the binary outcome (Hypothesis 1; Spearman correlation, rho = 0.31, $p$-value = 0.17) and the effect sizes (Hypothesis 2; Spearman correlation, rho = 0.26, $p$-value = 0.24). Similarly, for the survey-based forecast, we do not obtain a statistically significant correlation between forecasted and observed binary outcome (Hypothesis 3; Spearman correlation, rho = 0.14, $p$-value = 0.52), and between forecasted and observed effect size (Hypothesis 4; Spearman correlation, rho = 0.06, $p$-value = 0.78) (figure 1).

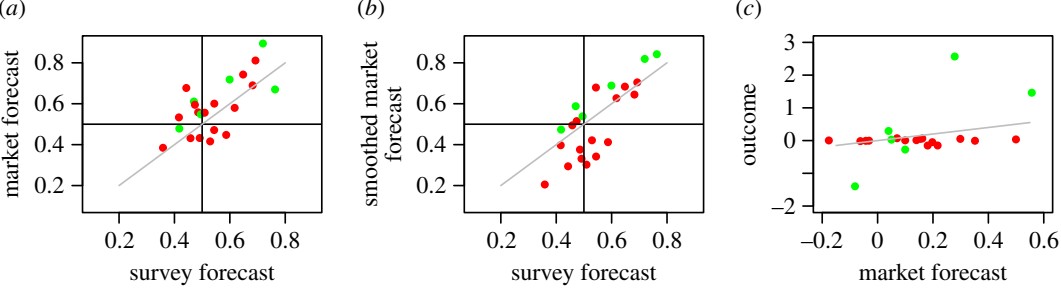

**Figure 1.** Forecasts and experimental outcomes. Green (red) dots indicate hypothesis that were supported (not supported) by the experimental data. The identity line is shown in grey. (*a*) Market forecasts versus survey-based forecasts for the binary outcomes. (*b*) Smoothed market forecasts versus survey-based forecasts for the binary outcomes. (*c*) Market forecasts versus observed outcomes for the effect sizes. Note that the observed effect sizes cover a much larger range than the forecasted effect sizes.

**Table 2.** Forecasts and observed outcomes.

| hypothesis | observed outcomes | | survey forecasts | | market forecasts | |
|---|---|---|---|---|---|---|
| | binary | Cohen's *d* | binary | Cohen's *d* | binary | Cohen's *d* |
| H1 | 0 | 0.01 | 0.62 | 0.28 | 0.58 | 0.14 |
| H2 | 1 | 0.30 | 0.60 | 0.27 | 0.72 | 0.04 |
| H3 | 1 | −1.40 | 0.72 | 0.18 | 0.89 | −0.08 |
| H4 | 0 | 0.07 | 0.54 | 0.13 | 0.60 | 0.07 |
| H5 | 0 | 0.07 | 0.54 | 0.18 | 0.47 | 0.16 |
| H6 | 1 | 0.03 | 0.42 | 0.01 | 0.48 | 0.05 |
| H7 | 0 | −0.02 | 0.42 | 0.08 | 0.53 | −0.06 |
| H8 | 0 | 0.05 | 0.36 | −0.01 | 0.38 | 0.15 |
| H9 | 0 | −0.05 | 0.68 | 0.32 | 0.69 | 0.20 |
| H10 | 0 | −0.15 | 0.53 | 0.17 | 0.42 | 0.18 |
| H11 | 1 | −0.27 | 0.47 | 0.11 | 0.61 | 0.10 |
| H12 | 0 | 0.01 | 0.59 | 0.15 | 0.45 | 0.10 |
| H13 | 0 | −0.01 | 0.65 | 0.24 | 0.74 | 0.35 |
| H14 | 0 | 0.00 | 0.46 | 0.12 | 0.43 | 0.80 |
| H15 | 0 | 0.01 | 0.47 | 0.07 | 0.60 | −0.18 |
| H16 | 0 | −0.01 | 0.49 | 0.06 | 0.56 | −0.04 |
| H17 | 0 | 0.04 | 0.69 | 0.29 | 0.81 | 0.50 |
| H18 | 1 | 2.57 | 0.50 | 0.13 | 0.55 | 0.28 |
| H19 | 0 | −0.15 | 0.49 | 0.14 | 0.43 | 0.22 |
| H20 | 0 | 0.05 | 0.44 | 0.00 | 0.68 | 0.30 |
| H21 | 0 | 0.00 | 0.51 | 0.08 | 0.56 | −0.03 |
| H22 | 1 | 1.46 | 0.76 | 0.44 | 0.67 | 0.56 |

When using absolute errors to compare the accuracy of market-based and survey-based forecasts, we find no evidence for a statistically significant difference for binary forecasts (median = 0.50 versus 0.51, Wilcoxon signed-rank test, $V = 123$, *p*-value = 0.92) and for effect size forecasts (median = 0.25 versus 0.13, $V = 124$, *p*-value = 0.95). Similarly, when using squared errors (Brier scores) to assess accuracy, we observe no evidence for a statistically significant difference for binary forecasts (median = 0.25 versus 0.26; Wilcoxon signed-rank test, $V = 133$, *p*-value = 0.85), or effect size forecasts (median = 0.06 versus 0.02; Wilcoxon signed-rank test, $V = 123$, *p*-value = 0.92).

The market forecasts tend to significantly overestimate observed binary outcomes (Wilcoxon signed-rank test, $V = 33$, $p$-value = 0.001), but not the observed effect sizes ($V = 78$, $p$-value = 0.12). There is suggestive evidence that the survey-based forecasts tend to significantly overestimate observed binary outcomes (Wilcoxon signed-rank test, $V = 49$, $p$-value = 0.01) and effect sizes (Wilcoxon signed-rank test, $V = 53$, $p$-value = 0.02).

The demographic variables (academia versus non-academia and expertise in the hypotheses; see Exploratory analyses) are not associated with survey predictions. Using individual absolute prediction error on survey predictions, in an ordinary least-squares individual data regression with clustering of standard errors on the individuals, we obtain regression coefficients of $-0.001$ ($t = -0.13$, $p$-value = 0.89) for expertise, and 0.03 ($t = 0.68$, $p$-value = 0.49) for being in academia.

In a robustness analysis, we use the average of the prices from the second half of the trading period (daily price snapshots) as alternative market forecasts. In this analysis, we observe suggestive evidence for a correlation for binary forecasts (rho = 0.48 $p$-value = 0.023) but not for effect size forecasts (rho = 0.004, $p$-value = 0.99. There is a statistically significant difference in accuracy between market- and survey-based forecasts for binary outcomes ($V = 35$, $p$-value = 0.002) but not for effect sizes ($V = 115$, $p$-value = 0.73) when using absolute error. Similar results are observed when using squared errors/Brier scores (suggestive evidence with $V = 43$, $p$-value = 0.005, and no evidence with $V = 114$, $p$-value = 0.70). Using the average of the prices from the second half of the trading period as alternative market forecasts, we observe suggestive evidence for overestimation (Wilcoxon signed-rank test, $V = 51$, $p$-value = 0.013) for binary outcomes but not for effect sizes (Wilcoxon signed-rank test, $V = 67$, $p$-value = 0.054). Overall, average prices from the second half of the trading period tend to thus provide better predictions than final market prices: the correlation with observed outcomes is stronger, and the Brier scores are statistically significantly different from the scores for the survey-based predictions for the binary outcomes. All analyses have been carried out in R (v. 3.5.1) [21]. For the clustering of standard errors in the exploratory analysis of the demographics questions, we used the package 'miceadds' [22].

# 4. Discussion

In this project, we find little evidence that researchers can predict outcomes of the hypotheses tested in NGS2. Whether this is due to the relatively small sample of hypotheses ($N = 22$), participants ($N = 39$) or the type of hypotheses tested is unclear. Here, unlike in most previous work, participants predicted tests from Bayesian analyses—whether this contributes to the poor performance of the markets and surveys is also unclear. An important difference compared with the previous prediction markets studies on direct replications is also that the original study $p$-value is an important predictor of replication outcomes, but such information is by definition not available for predicting the NGS2 outcomes, making it a more challenging prediction task for forecasters. Given the previously observed success in experts predicting novel outcomes with forecasting surveys (e.g. [20]), it may be the case that prediction markets function better for replication outcomes relative to forecasting surveys—more work on this topic would be needed for more definitive conclusions.

# 5. Deviations from protocol

We originally intended to predict the results from the 14 central hypotheses tested in the Cycle 2 NGS2 project by Diego-Rosell *et al.* [17] (also on capturing the effects of different types of uncertainty and perceived intergroup competition on the group motivation to innovate). In the original protocol (https://osf.io/uyaxk/), we had written that our participants would predict the outcomes of these 14 different hypotheses in terms of whether the result would be statistically significant with a $p$-value less than 0.05 or not. In the revised protocol, we instead have participants predict the outcomes of 22 hypotheses in Cycle 3 of the project and we ask them to predict whether the test result is in the same direction as the hypothesized one, with a Bayes factor of at least 10. The survey measures and prediction market measures have thus been updated accordingly, and in our work we still stick to null hypothesis significance testing. We also added monetary incentives to the survey—both the binary measure and the effect size measure are incentivized with a quadratic scoring rule. We have updated the introduction to make it clearer what the relationship between our work and Diego-Rosell *et al.* [14] is. In §2.1 Participants, we have also clarified that participants needed to have at least a MSc degree or to be working on related topics, that if we were to have less than 50 participants sign up we would not run the markets and that only the participants that fill out the survey are entitled to

participate in the markets. In the section on Statistical power, we have also clarified that previous work on which we base our power calculations is mainly on replication outcomes, unlike in the current project. Additionally, we have added a robustness test to the Secondary hypotheses (using the squared prediction error as a measure of prediction accuracy). We have also added three co-authors to the protocol (D.V., G.B. and P.D.-R.). Finally, we ran the prediction markets and surveys in October– November 2019. The approved Stage 1 manuscript, unchanged from the point of in-principle acceptance (23 January 2020) may be found at: https://osf.io/vgf79/.

Ethics. University of Virginia IRB approved the study (Ref: Project no. 2018019300). Participants give consent through an online form.

Data accessibility. Anonymized data and R scripts with the statistical analysis are publicly available through OSF: https:// osf.io/skjqm/. Materials (including information to participants and consent forms) are also on OSF: https://osf.io/ 8p2ac/, including the original protocol.

Authors' contributions. Y.C., M.J., B.A.N., T.P., A.S. and A.D. conceived of the study, participated in the design of the study, coordinated the study and drafted the manuscript. D.V. participated in the design of the study, coordinated the study and drafted the manuscript. G.B. and P.D.-R. participated in the design of the study and drafted the manuscript. T.P. and P.D.-R. carried out the statistical analyses. All authors gave final approval for publication.

Competing interests. Cultivate Labs employs A.S. and provides the online market interface used in the experiment. The market interface is commercial software. B.A.N. is an employee of the Center for Open Science, a non-profit technology and culture change organization with a mission to increase openness, integrity, and reproducibility of research. G.B. and P.D.-R. are employees at Gallup, an analytics and advisory company. The other authors have no competing interests.

Funding. A.D. is funded by the Knut and Alice Wallenberg Foundation and the Marcus and Marianne Wallenberg Foundation (through a Wallenberg Scholar grant), the Austrian Science Fund (FWF, SFB F63) and the Jan Wallander and Tom Hedelius Foundation (Svenska Handelsbankens Forskningsstiftelser) and A.D. and M.J. are funded by the Swedish Foundation for Humanities and Social Sciences. T.P. is funded by Marsden Fund grant nos. 16-UOA-190 and 17-MAU-133.

Acknowledgements. The authors thank Alex DeHaven and Brandon Thorpe for all their help with this project.

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
