## [Peer Review File · Royal Society Open Science]

Review History

Decision letter (RSOS-181036.R0)

02-Aug-2018

Dear Dr Dreber,

I write you in regards to manuscript RSOS-181036 entitled "Using prediction markets to address the incoherency problem in social science" which you submitted to Royal Society Open Science.

We routinely triage submissions for scientific soundness, clarity and general adherence to the Registered Reports guidelines. For submissions that have promise but are not yet suitable for in-depth Stage 1 review, we offer feedback to help authors maximise the chances that reviewers will respond positively to a resubmission.

We have concluded that your submission is not yet suitable for in-depth review and has therefore been rejected at this time, but we believe it will be suitable once several issues are addressed. We

therefore invite a resubmission. Further comments from the Associate Editor may be found at the end of this letter.

If you wish to revise your manuscript in light of the below comments please submit your manuscript as a new submission and mention this previous manuscript ID in your covering letter. You should also provide a detailed response to the below comments in the cover letter.

Thank you for considering Royal Society Open Science for the publication of your registered report.

Kind regards,
Thadcha Retneswaran
Royal Society Open Science Editorial Office
Royal Society Open Science
openscience@royalsociety.org

on behalf of Chris Chambers (Registered Reports Editor, Royal Society Open Science)
openscience@royalsociety.org

Associate Editor Comments to Author:

Before we can proceed with in-depth review, please attend to the following points:

1. Include a formal power analysis (or appropriate alternative), including justification for the target effect size. RSOS sets no minimum level of statistical power but a stronger rationale for the chosen effect size (and a minimum N) are required.
2. The detailed description of the hypotheses and analysis plans is good, but I would recommend including subheadings above each pair of null and alternative hypotheses to make clear to reviewers the question each hypothesis is addressing.
3. The Discussion section should be removed - it is not necessary at Stage 1 and would need to be replaced at Stage 2 anyway. Only Introduction and Methods should be included in Stage 1 submissions.
4. Are there any inclusion criteria for participation? And are there any conditions under which you would discard data, either discarding an entire participant or a subset of their data? Please ensure that any such conditions are fully specified.
5. Does the appropriateness of the proposed statistical tests depend on any informal assumptions, e.g. what if the relationship between variables is non-monotonic? Would Spearman's correlations still be appropriate? What contingencies, if any, are in place for violation of those assumptions?

Author's Response to Decision Letter for (RSOS-181036.R0)

Dear Prof Chambers,

Thanks for your comments and we have now addressed the five points. We include a formal power analysis, we include subheadings above each pair of null and alternative hypotheses, we have removed the Discussion section, we explicitly discuss inclusion criteria for participation, and we discuss why we use Spearman's correlations.

Best wishes and thanks!
Anna (on behalf of all coauthors)

RSOS-181308.R0

Review form: Reviewer 1

Is the language acceptable?

Yes

Do you have any ethical concerns with this paper?

No

Have you any concerns about statistical analyses in this paper?

Yes

Recommendation?

Accept with minor revision

Comments to the Author(s)

Please see attached file (Appendix A).

Review form: Reviewer 2

Is the language acceptable?

Yes

Do you have any ethical concerns with this paper?

No

Have you any concerns about statistical analyses in this paper?

No

Recommendation?

Accept with minor revision

Comments to the Author(s)

Review for "Using prediction markets to address the incoherency problem in social science"

Liangfei Qiu
University of Florida

This research proposes to examine a timely and important question: can prediction markets address the incoherency problem in social science? This research question has potential to contribute significantly to the prediction market literature and has important policy implications for academic research.

The proposed hypotheses are interesting and reasonable. In this proposal, the authors want to test if experts are better than random in predicting the results. I was wondering if another related hypothesis can also be tested: whether experts are better than normal people (e.g. college students) in predicting the results. In particular, the underlying mechanism that can effectively incentivize experts vs. normal people may be different.

The methodology of the proposal is rigorous. The authors are very careful about interpreting their results. The description of the methods is clear and easy to follow. The authors provide sufficient details for readers to understand the research design. About research design, I was wondering if the monetary incentive (less than 100 dollars) is sufficient for the experts to provide thoughtful responses (these experts are not college students in typical laboratory experiments in economics and psychology)? Using non-pecuniary incentives may improve individual prediction accuracy in social-media based prediction markets. In other words, social effects and reputational concerns can play a key role in improving participants' prediction accuracy, and provide a powerful motivation for prediction market participants to supply thoughtful responses. If the participants (experts) are encouraged to share their predictions on social media, they may provide more thoughtful predictions.

Overall, I enjoyed reading this report and would like to recommend "Accept with minor revision."

Review form: Reviewer 3

Is the language acceptable?

Yes

Do you have any ethical concerns with this paper?

No

Have you any concerns about statistical analyses in this paper?

No

Recommendation?

Accept with minor revision

Comments to the Author(s)

Manuscript ID: RSOS-181308.

Title: Using prediction markets to address the incoherency problem in social science.

Authors: Anna Dreber, Yiling Chen, Magnus Johannesson, Brian Nosek, Thomas Pfeiffer and Adam Siegel.

Overall Assessment:

1. The registered report addresses an important research question, namely, "Could which research hypothesis receives empirical support or not be predicted ex-ante?" It is also quite topical given the surge of attention both inside and outside the scientific community about the ways in which the knowledge system works.
2. The authors repeatedly invite a judgement on the contribution of their proposed study in conjunction with the data they have collected in previous studies. While there is definitely value in investigating similar research questions to previous studies in a different domain, I

would strongly invite the authors to enhance the innovative contribution of the proposed study by investigating other aspects, which were not into the focus of previous studies and/or may make the inter-study comparisons/integrations more informative.

3. The hypotheses stated seem plausible though they could be made more sophisticated (see below). Similarly, their rationale could be detailed more.

4. The methodology is overall sound. However, given the study design, I would be concerned ex-ante about possible order effects between the survey and the prediction markets that may affect the interpretation of any correlation between survey and market outcomes. I would recommend the author to make appropriate design changes to control for order effects.

5. In general, the details of (and sometimes the motivations for) the study methodology are lacking in this report (though they could be inferred from some of the studies referenced).

Incidentally, it would have been useful to include supplemental material to provide more details about the study methodology.

6. Hopefully, the authors will find my commentary useful and may be able to improve further their proposed study and deliver interesting findings for the scientific community.

Major Comments:

7. Both the title and the abstract do not seem to reflect fully the main body of the report (e.g., reference to the "incoherence problem").

8. The introduction – rightly so – puts at centre stage the DARPA's Next Generation Social Science (NGS2) programme. However, I fear that for readers without prior familiarity with the programme the information provided by the authors is too limited and somewhat airy to understand fully the context of the proposed study. I feel the authors should make an additional effort to communicate the core elements of the programme and provide the reader with more details either in a footnote or in an appendix (or both).

9. The final paragraph of the introduction defines a hypothesis as 'supported' by relying on a p-value less than 0.005. It is unclear why this should be the relevant threshold. Why not the canonical 0.05? I think the authors should motivate their choice of a threshold in detail by also arguing as much as possible how such a choice will not impair the inference that can be drawn by the study.

10. The authors intend to recruit participants via Brian Nosek's twitter feed. This recruitment strategy may allow reaching a wider audience, but it does not allow precise participant targeting. (I guess non-researchers may also take part in the study). Therefore, while it is fine not to exclude any participant/discard any data, it would be insightful to learn about participants' types. Indeed, ex-ante I would expect different predictions coming from either researchers or lay people; and, among researchers, different predictions coming from those closer to the field of investigation of the hypotheses being tested relatively to those coming from researchers further away. I would invite the authors to elicit background characteristics, which may allow them to classify participants in types relevant to the prediction exercise.

11. The lottery incentive system for the survey measures seems – in expected terms – quite bland. Moreover, some assessment of belief elicitation methods points towards the direction of mixed effects in using incentives when eliciting beliefs (e.g., Schotter and Trevino, 2014). Why not running the survey without monetary incentives? Would not be best to not pay at all (perhaps, just keeping a flat fee for participation) if paying enough is not feasible (e.g., Gneezy and Rustichini, 2000)? After all, the prediction market should be the mechanism where monetary incentives are at play to elicit beliefs in an incentive-compatible way.

12. The prediction markets will be open for two weeks. However, it could happen that in such a time window only few people will volunteer to take part in the study. The authors should commit to a minimum sample size to consider the study as successful implemented for leading to a Royal Society Open Science publication. They should also argue why such a sample size is the right as minimum sample size and what alternative plans they will put in place in case the minimum sample size is not met within the two-week window.

13. Not all participants may easily understand how prediction market and its web-based interface work. I would recommend the authors to build in opportunities for the participants to familiarise with the functioning of the market and its interface as well as ways of testing their understanding. Any successful attempt in this direction may reduce noise in the data and enhance the robustness of the interpretation of the empirical evidence.

14. It seems that the benchmark for hypothesis testing of a random prediction is a too easy target and not a very informative one. I would urge the authors to elaborate further (for instance on a 'scientifically significant' difference relatively to a random prediction) and formulate more challenging and informative benchmarks.

15. Integrate the classical hypothesis testing with alternative approaches and/or econometric analysis would also constitute added value.

Minor Comments:

16. Page 3, row 20: the acronym OSF appears without being introduced.

17. Page 4, row 23: I would invite the authors to use a different letter for the binary variable rather than p^* : indeed, p is a letter usually used to indicate probabilities and employing it for the binary variable may generate unnecessary, though frankly limited, confusion.

18. Page 4, row 51: there is a typo "maker" should read "market".

References

Gneezy, Uri, and Aldo Rustichini. 2000. Pay Enough or Don't Pay at All. *The Quarterly Journal of Economics*, 115 (3), 791–810.

Schotter, Andrew, and Isabel Trevino. 2014. Belief Elicitation in the Laboratory. *Annual Review of Economics*, 6(1), 103-128.

Decision letter (RSOS-181308.R0)

17-Sep-2018

Dear Dr Dreber,

The Editors assigned to your Stage 1 Registered Report ("Using prediction markets to address the incoherency problem in social science") have now received comments from reviewers. We would like you to revise your paper in accordance with the referee and editors suggestions which can be found below (not including confidential reports to the Editor). Please note this decision does not guarantee eventual acceptance.

Please note that Royal Society Open Science charge article processing charges for all new submissions that are accepted for publication. Charges will also apply to papers transferred to

Royal Society Open Science from other Royal Society Publishing journals, as well as papers submitted as part of our collaboration with the Royal Society of Chemistry (<http://rsos.royalsocietypublishing.org/chemistry>). If your manuscript is newly submitted and subsequently accepted for publication, you will be asked to pay the article processing charge, unless you request a waiver and this is approved by Royal Society Publishing. You can find out more about the charges at <http://rsos.royalsocietypublishing.org/page/charges>. Should you have any queries, please contact openscience@royalsociety.org.

on behalf of Chris Chambers (Registered Reports Editor, Royal Society Open Science)
openscience@royalsociety.org

Associate Editor Comments to Author (Professor Chris Chambers):

Associate Editor: 1

Comments to the Author:

Three expert reviewers have now assessed the submission. All find various degrees of merit in the proposal, while also offering a wide range of constructive suggestions for improvement. To highlight some of the key points: Reviewer 1 questions the necessarily low sample size and proposes a data aggregation strategy to increase the value of the research, together with the removal of all inferential statistical tests. From an editorial point of view, I am open to arguments either way in favour of significance testing in this context, but would invite the authors to clearly justify their approach. Reviewer 2 is positive but questions the level of financial incentive and, moreover, whether the design is sufficiently calibrated to maximise non-pecuniary incentives. This latter point is echoed by Reviewer 3 who is the most critical and, among wide range of concerns, also asks whether the research question is sufficiently substantive, whether the design accounts for order effects, and whether the sampling method is sufficiently informative (eg. whether further classification of participants is required). The reviewer also challenges the justification of the proposed hypotheses.

Overall I am convinced by the reviews that there is sufficient merit in the proposal to invite a substantial revision. Depending on the comprehensiveness of the response to reviewers, the manuscript may be returned to one or more of the reviewers for reassessment.

Comments to Author:

Reviewer: 1

Comments to the Author(s)

Please see attached file

Reviewer: 2

Comments to the Author(s)

Review for "Using prediction markets to address the incoherency problem in social science"

Liangfei Qiu
University of Florida

This research proposes to examine a timely and important question: can prediction markets address the incoherency problem in social science? This research question has potential to contribute significantly to the prediction market literature and has important policy implications for academic research.

The proposed hypotheses are interesting and reasonable. In this proposal, the authors want to test if experts are better than random in predicting the results. I was wondering if another related hypothesis can also be tested: whether experts are better than normal people (e.g. college students) in predicting the results. In particular, the underlying mechanism that can effectively incentivize experts vs. normal people may be different.

The methodology of the proposal is rigorous. The authors are very careful about interpreting their results. The description of the methods is clear and easy to follow. The authors provide sufficient details for readers to understand the research design. About research design, I was wondering if the monetary incentive (less than 100 dollars) is sufficient for the experts to provide thoughtful responses (these experts are not college students in typical laboratory experiments in economics and psychology)? Using non-pecuniary incentives may improve individual prediction accuracy in social-media based prediction markets. In other words, social effects and reputational concerns can play a key role in improving participants' prediction accuracy, and provide a powerful motivation for prediction market participants to supply thoughtful responses. If the participants (experts) are encouraged to share their predictions on social media, they may provide more thoughtful predictions.

Overall, I enjoyed reading this report and would like to recommend "Accept with minor revision."

Reviewer: 3

Comments to the Author(s)

Manuscript ID: RSOS-181308.

Title: Using prediction markets to address the incoherency problem in social science.

Authors: Anna Dreber, Yiling Chen, Magnus Johannesson, Brian Nosek, Thomas Pfeiffer and Adam Siegel.

Overall Assessment:

1. The registered report addresses an important research question, namely, "Could which research hypothesis receives empirical support or not be predicted ex-ante?" It is also quite topical given the surge of attention both inside and outside the scientific community about the ways in which the knowledge system works.
2. The authors repeatedly invite a judgement on the contribution of their proposed study in conjunction with the data they have collected in previous studies. While there is definitely value in investigating similar research questions to previous studies in a different domain, I would strongly invite the authors to enhance the innovative contribution of the proposed study by investigating other aspects, which were not into the focus of previous studies and/or may make the inter-study comparisons/integrations more informative.
3. The hypotheses stated seem plausible though they could be made more sophisticated (see below). Similarly, their rationale could be detailed more.
4. The methodology is overall sound. However, given the study design, I would be concerned ex-ante about possible order effects between the survey and the prediction markets that may affect the interpretation of any correlation between survey and market outcomes. I would recommend the author to make appropriate design changes to control for order effects.

5. In general, the details of (and sometimes the motivations for) the study methodology are lacking in this report (though they could be inferred from some of the studies referenced). Incidentally, it would have been useful to include supplemental material to provide more details about the study methodology.
6. Hopefully, the authors will find my commentary useful and may be able to improve further their proposed study and deliver interesting findings for the scientific community.

Major Comments:

7. Both the title and the abstract do not seem to reflect fully the main body of the report (e.g., reference to the "incoherence problem").
8. The introduction – rightly so – puts at centre stage the DARPA's Next Generation Social Science (NGS2) programme. However, I fear that for readers without prior familiarity with the programme the information provided by the authors is too limited and somewhat airy to understand fully the context of the proposed study. I feel the authors should make an additional effort to communicate the core elements of the programme and provide the reader with more details either in a footnote or in an appendix (or both).
9. The final paragraph of the introduction defines a hypothesis as 'supported' by relying on a p-value less than 0.005. It is unclear why this should be the relevant threshold. Why not the canonical 0.05? I think the authors should motivate their choice of a threshold in detail by also arguing as much as possible how such a choice will not impair the inference that can be drawn by the study.
10. The authors intend to recruit participants via Brian Nosek's twitter feed. This recruitment strategy may allow reaching a wider audience, but it does not allow precise participant targeting. (I guess non-researchers may also take part in the study). Therefore, while it is fine not to exclude any participant/discard any data, it would be insightful to learn about participants' types. Indeed, ex-ante I would expect different predictions coming from either researchers or lay people; and, among researchers, different predictions coming from those closer to the field of investigation of the hypotheses being tested relatively to those coming from researchers further away. I would invite the authors to elicit background characteristics, which may allow them to classify participants in types relevant to the prediction exercise.
11. The lottery incentive system for the survey measures seems – in expected terms – quite bland. Moreover, some assessment of belief elicitation methods points towards the direction of mixed effects in using incentives when eliciting beliefs (e.g., Schotter and Trevino, 2014). Why not running the survey without monetary incentives? Would not be best to not pay at all (perhaps, just keeping a flat fee for participation) if paying enough is not feasible (e.g., Gneezy and Rustichini, 2000)? After all, the prediction market should be the mechanism where monetary incentives are at play to elicit beliefs in an incentive-compatible way.
12. The prediction markets will be open for two weeks. However, it could happen that in such a time window only few people will volunteer to take part in the study. The authors should commit to a minimum sample size to consider the study as successfully implemented for leading to a Royal Society Open Science publication. They should also argue why such a sample size is the right as minimum sample size and what alternative plans they will put in place in case the minimum sample size is not met within the two-week window.
13. Not all participants may easily understand how prediction market and its web-based interface work. I would recommend the authors to build in opportunities for the participants to familiarise with the functioning of the market and its interface as well as ways of testing their understanding. Any successful attempt in this direction may reduce noise in the data and enhance the robustness of the interpretation of the empirical evidence.
14. It seems that the benchmark for hypothesis testing of a random prediction is a too easy target and not a very informative one. I would urge the authors to elaborate further (for instance on a 'scientifically significant' difference relatively to a random prediction) and formulate more challenging and informative benchmarks.

15. Integrate the classical hypothesis testing with alternative approaches and/or econometric analysis would also constitute added value.

Minor Comments:

16. Page 3, row 20: the acronym OSF appears without being introduced.

17. Page 4, row 23: I would invite the authors to use a different letter for the binary variable rather than p^* : indeed, p is a letter usually used to indicate probabilities and employing it for the binary variable may generate unnecessary, though frankly limited, confusion.

18. Page 4, row 51: there is a typo “maker” should read “market”.

References

Gneezy, Uri, and Aldo Rustichini. 2000. Pay Enough or Don't Pay at All. *The Quarterly Journal of Economics*, 115 (3), 791–810.

Schotter, Andrew, and Isabel Trevino. 2014. Belief Elicitation in the Laboratory. *Annual Review of Economics*, 6(1), 103-128.

Author's Response to Decision Letter for (RSOS-181308.R0)

See Appendix B.

RSOS-181308.R1 (Revision)

Review form: Reviewer 1

Is the language acceptable?

Yes

Do you have any ethical concerns with this paper?

No

Have you any concerns about statistical analyses in this paper?

No

Recommendation?

Accept in principle

Comments to the Author(s)

The authors have responded satisfactorily to the points raised in the first round of review. I am happy to recommend that the manuscript be accepted in principle.

Review form: Reviewer 2

Is the language acceptable?

Yes

Do you have any ethical concerns with this paper?

No

Have you any concerns about statistical analyses in this paper?

No

Recommendation?

Accept in principle

Comments to the Author(s)

Review for "Using prediction markets to predict the outcomes in DARPA's Next Generation Social Science program"

Liangfei Qiu
University of Florida

In this comprehensive revision, the authors have addressed all my comments in the previous round. In particular, the authors ask their participants for demographic information that will be analyzed in an exploratory analysis. Therefore, I would like to recommend "Accept in principle."³

Review form: Reviewer 3

Is the language acceptable?

Yes

Do you have any ethical concerns with this paper?

No

Have you any concerns about statistical analyses in this paper?

No

Recommendation?

Reject

Comments to the Author(s)

Royal Society Open Science
Manuscript ID: RSOS-181308.R1.

Title: Using prediction markets to predict the outcomes in DARPA's Next Generation Social Science program.

Authors: Anna Dreber, Yiling Chen, Magnus Johannesson, Brian Nosek, Thomas Pfeiffer and Adam Siegel.

Comments:

1. I praise the authors for tackling some of my previous comments. However, important concerns were not substantially addressed. A main concern remains the plea by the authors to judge the contribution of their proposed study in conjunction with the data they have collected in previous studies (or will be collected as part of their research programme). This is made in particular with reference to the relatively small sample size (see authors' response to my comment 12) and

consequent study power. Focusing only on the proposed study under review, I find these to be severe limitations.

2. Those limitations have important repercussions on other issues as, for instance, the lack of both more sophisticated hypothesis (see my comments no. 3 and 14) and controls for order effects (see my major comment no. 4); any learning from eliciting participants' types will also be hindered (see my comment no. 10) as well as the use of alternative statistical/econometric approaches (see my comment no. 15).

3. The introduction is still somewhat airy and it is not easy to grasp the key information (see my comment no. 8). This is especially true -- in my view -- for the first two paragraphs, which were not altered (as far as I was able to detect) when revising the manuscript. I would suggest the authors to significantly rewrite (at least) those paragraphs to make them sharper, more crystal-clear and self-contained.

4. The authors could do more in motivating their choice of a statistical threshold (see my comment no. 9) especially by (but not limited to) also elaborating on the impact it may have on survey responses and market predictions.

5. I would also strongly invite the authors to put in place ways to ensure participants' understanding of prediction markets and collect relevant evidence showing that this is indeed the case. I think that relying on past experiences that "seem to work well" is not optimal.

6. In future, I would recommend the authors to explicitly mention in their response to reviewers how the manuscript has been revised to enhance comparability with the previous submitted version.

7. Once again, I hope the authors will find my commentary useful and may be able to improve further their proposed study and deliver interesting findings for the scientific community.

Decision letter (RSOS-181308.R1)

19-Dec-2018

Dear Dr Dreber,

The Subject Editor assigned to your paper ("Using prediction markets to predict the outcomes in DARPA's Next Generation Social Science program") has now received comments from reviewers. We would like you to revise your paper in accordance with the referee and Associate Editor suggestions which can be found below (not including confidential reports to the Editor).

Please note that Royal Society Open Science charge article processing charges for all new submissions that are accepted for publication. Charges will also apply to papers transferred to Royal Society Open Science from other Royal Society Publishing journals, as well as papers submitted as part of our collaboration with the Royal Society of Chemistry

(<http://rsos.royalsocietypublishing.org/chemistry>). If your manuscript is newly submitted and subsequently accepted for publication, you will be asked to pay the article processing charge, unless you request a waiver and this is approved by Royal Society Publishing. You can find out more about the charges at <http://rsos.royalsocietypublishing.org/page/charges>. Should you have any queries, please contact openscience@royalsociety.org.

on behalf of Professor Chris Chambers (Subject Editor)
openscience@royalsociety.org

Editor Comments to Author (Professor Chris Chambers):

The revised manuscript was reassessed by the three original reviewers. Reviewers 1 and 2 are now satisfied and recommend IPA; however, Reviewer 3 is unsatisfied with several responses and recommends Rejection. Based on the appraisals of Reviewer 1 and 2 (as well as my own reading of the manuscript), I believe a Reject decision is too severe. That said, I would like to see how the authors respond to the remaining objections of Reviewer 3, which do appear to have merit, especially concerns about sample size, order effects and clarity of the Introduction (comments 1, 2, 3 and 5). A final editorial decision will be made swiftly on the basis of the authors' response and any corresponding final revisions, most likely without requiring further in-depth Stage 1 review.

Reviewer comments to Author:
Reviewer: 1

Comments to the Author(s)
The authors have responded satisfactorily to the points raised in the first round of review. I am happy to recommend that the manuscript be accepted in principle.

Reviewer: 2
Comments to the Author(s)
Review for "Using prediction markets to predict the outcomes in DARPA's Next Generation Social Science program"

Liangfei Qiu
University of Florida

In this comprehensive revision, the authors have addressed all my comments in the previous round. In particular, the authors ask their participants for demographic information that will be analyzed in an exploratory analysis. Therefore, I would like to recommend "Accept in principle."

Reviewer: 3
Comments to the Author(s)
Royal Society Open Science
Manuscript ID: RSOS-181308.R1.

Title: Using prediction markets to predict the outcomes in DARPA's Next Generation Social Science program.

Authors: Anna Dreber, Yiling Chen, Magnus Johannesson, Brian Nosek, Thomas Pfeiffer and Adam Siegel.

Comments:

1. I praise the authors for tackling some of my previous comments. However, important concerns were not substantially addressed. A main concern remains the plea by the authors to judge the contribution of their proposed study in conjunction with the data they have collected in previous studies (or will be collected as part of their research programme). This is made in particular with reference to the relatively small sample size (see authors' response to my comment 12) and consequent study power. Focusing only on the proposed study under review, I find these to be severe limitations.
2. Those limitations have important repercussions on other issues as, for instance, the lack of both more sophisticated hypothesis (see my comments no. 3 and 14) and controls for order effects (see my major comment no. 4); any learning from eliciting participants' types will also be hindered (see my comment no. 10) as well as the use of alternative statistical/econometric approaches (see my comment no. 15).
3. The introduction is still somewhat airy and it is not easy to grasp the key information (see my comment no. 8). This is especially true -- in my view -- for the first two paragraphs, which were not altered (as far as I was able to detect) when revising the manuscript. I would suggest the authors to significantly rewrite (at least) those paragraphs to make them sharper, more crystal-clear and self-contained.
4. The authors could do more in motivating their choice of a statistical threshold (see my comment no. 9) especially by (but not limited to) also elaborating on the impact it may have on survey responses and market predictions.
5. I would also strongly invite the authors to put in place ways to ensure participants' understanding of prediction markets and collect relevant evidence showing that this is indeed the case. I think that relying on past experiences that "seem to work well" is not optimal.
6. In future, I would recommend the authors to explicitly mention in their response to reviewers how the manuscript has been revised to enhance comparability with the previous submitted version.
7. Once again, I hope the authors will find my commentary useful and may be able to improve further their proposed study and deliver interesting findings for the scientific community.

Author's Response to Decision Letter for (RSOS-181308.R1)

See Appendix C.

RSOS-181308.R2 (Revision)

Review form: Reviewer 1

Do you have any ethical concerns with this paper?

No

Recommendation?

Reject

Comments to the Author(s)

Thank you for the opportunity to re-review this manuscript. After stage 1 review and in-principle acceptance, the study has changed substantially. The authors provide clear and compelling explanations and motivations for the changes. However, a preregistration was made on OSF, and data collection is now completed under that preregistration. This means that any further requests for revisions at stage 1 will not be able to influence data collection, and may conflict with the authors' preregistration. Therefore, I question whether this manuscript should be re-reviewed at stage 1, rather than either moving to stage 2 or withdrawing the registration. I remain convinced that this work makes a valuable contribution, and I look forward to the editor's appraisal. Since a recommendation is required, I have made a pro forma entry of "reject", based on the procedural concerns outlined above and not on the substantive content of the manuscript.

Review form: Reviewer 2**Do you have any ethical concerns with this paper?**

No

Recommendation?

See Appendix C.

Comments to the Author(s)

I have reviewed the previous version of the paper. It studies an interesting and timely research question. The paper is written well, and the research method is rigorous. I have the following comments, and I hope they are helpful.

1. Participants will be recruited through social media via Brian Nosek's twitter feed and other behavioral science organizations. Why the participants will be recruited using a particular twitter feed (instead of sending emails to a given set of researchers)? The authors may want to justify it.
2. Potentially, the authors could investigate how demographic information affects the hypotheses. In particular, I am interested in how the participants' expertise affects the absolute prediction error between the prediction market and the average survey. In other words, does the advantage of prediction market relative to survey depend on the participants' expertise?

Review form: Reviewer 3**Do you have any ethical concerns with this paper?**

No

Recommendation?

Major revision

Comments to the Author(s)

1. A main concern remains the relatively small sample size. A minimal target sample of 50 participants is too low and, consequently, the study should be regarded as very speculative. The

language should be adjusted accordingly. Similarly, the statistical analysis should prominently address concerns due to a small number of independent observations while testing predictions about a relative large number of hypotheses (which now totalise to 22, with an increase of 57% relatively to the last version of this report).

2. Related to sample size, there is the issue of study power. In the current version, the estimated power has substantially increased with respect to the previous version of this report. However, it is unclear why this is the case, given that there is no substantial change in the observed correlations and the target sample size. Could it be an “artefact” of the increase in the number (from 14 to 22) of hypotheses? Or of the adoption of “Bayes factor”? Furthermore, the power analysis are made on the basis of prediction markets in the context of replications. This latter is arguably a different context from predicting new scientific results. Once again, the analysis should be taken as very speculative.

3. The introduction refers to Diego-Rosell et al. but it does not provide sufficient details to understand what they do and what the relationship is between their work and this study. The authors should rewrite (at least) the relevant paragraphs to make them sharper, more crystal-clear and self-contained.

4. It is unclear why the authors have made the shift from null hypothesis significance testing (p-value less than 0.05) to Bayes factor (of at least 10) to assess if a given hypothesis is supported. Arguably, Bayes factors are much less common in the statistical practices and, therefore, study participants may more likely be unfamiliar with them. Even more so, if we consider that the minimal requirement to be categorised as a researcher is to have a master degree or to be working on related topics. All of that again may be a factor hindering the robustness of the findings.

5. The incentives for participants appear quite bland. They are set in such a way that what matters is the average prediction made by participants. This is at odd with the hypothesis put forward in this report that focus on the individual predictions. It seems therefore that there is an issue of incentive compatibility.

6. The questions about researchers’ academic background could be improved by making the categories better defined (for instance, considering possible differences across disciplines and/or the US system and beyond). Incidentally, the inclusion of students in dummy variable of the exploratory analyses seems to hint that the authors expect a significant number of researchers participating in the study to be actually student. If this would be the case, I guess it would be yet another instance of a factor hindering the robustness of the findings.

7. Responses to past commentaries were not provided and many past comments seemed to have been overlooked. Similarly, no rationale for the changes made was offered making the referee work more of a guessing game. The authors should take the time to address past comments and explicitly provide their response to reviewers’ commentaries as well as how and – especially – why the manuscript has been revised.

Decision letter (RSOS-181308.R2)

06-Feb-2019

Dear Dr Dreber

On behalf of the Editor, I am pleased to inform you that your Stage 1 Registered Report RSOS-

181308.R2 entitled "Using prediction markets to predict the outcomes in DARPA's Next Generation Social Science program" has been accepted in principle for publication in Royal Society Open Science.

You may now progress to Stage 2 and complete the study as approved. Before commencing data collection we ask that you:

- 1) Update the journal office as to the anticipated completion date of your study.
- 2) Register your approved protocol on the Open Science Framework (<https://osf.io/>) or other recognised repository, either publicly or privately under embargo until submission of the Stage 2 manuscript. Please note that a time-stamped, independent registration of the protocol is mandatory under journal policy, and manuscripts that do not conform to this requirement cannot be considered at Stage 2. The protocol should be registered unchanged from its current approved state, with the time-stamp preceding implementation of the approved study design. We strongly recommend using the dedicated Registered Reports portal at <https://osf.io/rr>

Following completion of your study, we invite you to resubmit your paper for peer review as a Stage 2 Registered Report. Please note that your manuscript can still be rejected for publication at Stage 2 if the Editors consider any of the following conditions to be met:

- The results were unable to test the authors' proposed hypotheses by failing to meet the approved outcome-neutral criteria.
- The authors altered the Introduction, rationale, or hypotheses, as approved in the Stage 1 submission.
- The authors failed to adhere closely to the registered experimental procedures. Please note that any deviations from the approved experimental procedures must be communicated to the editor immediately for approval, and prior to the completion of data collection. Failure to do so can result in revocation of in-principle acceptance and rejection at Stage 2 (see complete guidelines for further information).
- Any post-hoc (unregistered) analyses were either unjustified, insufficiently caveated, or overly dominant in shaping the authors' conclusions.
- The authors' conclusions were not justified given the data obtained.

We encourage you to read the complete guidelines for authors concerning Stage 2 submissions at <http://rsos.royalsocietypublishing.org/content/registered-reports>. Please especially note the requirements for data sharing, reporting the URL of the independently registered protocol, and that withdrawing your manuscript will result in publication of a Withdrawn Registration.

Please note that Royal Society Open Science will introduce article processing charges for all new submissions received from 1 January 2018. Registered Reports submitted and accepted after this date will ONLY be subject to a charge if they subsequently progress to and are accepted as Stage 2 Registered Reports. If your manuscript is submitted and accepted for publication after 1 January 2018 (i.e. as a full Stage 2 Registered Report), you will be asked to pay the article processing charge, unless you request a waiver and this is approved by Royal Society Publishing. You can find out more about the charges at <http://rsos.royalsocietypublishing.org/page/charges>. Should you have any queries, please contact openscience@royalsociety.org.

Once again, thank you for submitting your manuscript to Royal Society Open Science and we look forward to receiving your Stage 2 submission. If you have any questions at all, please do not hesitate to get in touch. We look forward to hearing from you shortly with the anticipated submission date for your stage two manuscript.

on behalf of Professor Chris Chambers (Registered Reports Editor, Royal Society Open Science)
openscience@royalsociety.org

Associate Editor Comments to Author (Professor Chris Chambers):

This is an unusual situation involving the reassessment of an amended Stage 1 protocol following IPA. The three original reviewers who assessed the original Stage 1 manuscript have now completed their reassessments. Reviewers 1 and 2 are broadly positive, although Reviewer 1 recommends rejection on essentially administrative grounds. The reviewer makes an important point about the the amended protocol potentially conflicting with the existing preregistration (which is an issue that I can see confusing readers), but rather than issuing a procedural rejection, I would like to offer the authors the opportunity to anticipate and respond to this concern in the manuscript using a 'Deviations from Protocol' section, ensuring that the changes are comprehensively explained and justified. Reviewer 2 is similarly positive but offers some minor recommendations for revision, including stronger justification of the recruitment mode. Reviewer 3 remains negatively inclined, and although I was sufficiently convinced by the authors' previous responses to offer IPA, the reactivation of the Stage 1 review process provides the opportunity to address these concerns (and the reviewer's additional reasonable concerns) more adequately than in the previous version of the manuscript. Based on this combination of reviews, a Major Revision is recommended.

Reviewer Comments to Author:

Reviewer: 1

Comments to the Author(s)

Thank you for the opportunity to re-review this manuscript. After stage 1 review and in-principle acceptance, the study has changed substantially. The authors provide clear and compelling explanations and motivations for the changes. However, a preregistration was made on OSF, and data collection is now completed under that preregistration. This means that any further requests for revisions at stage 1 will not be able to influence data collection, and may conflict with the authors' preregistration. Therefore, I question whether this manuscript should be re-reviewed at stage 1, rather than either moving to stage 2 or withdrawing the registration. I remain convinced that this work makes a valuable contribution, and I look forward to the editor's appraisal. Since a recommendation is required, I have made a pro forma entry of "reject", based on the procedural concerns outlined above and not on the substantive content of the manuscript.

Reviewer: 2

Comments to the Author(s)

I have reviewed the previous version of the paper. It studies an interesting and timely research question. The paper is written well, and the research method is rigorous. I have the following comments, and I hope they are helpful.

1. Participants will be recruited through social media via Brian Nosek's twitter feed and

other behavioral science organizations. Why the participants will be recruited using a particular twitter feed (instead of sending emails to a given set of researchers)? The authors may want to justify it.

2. Potentially, the authors could investigate how demographic information affects the hypotheses. In particular, I am interested in how the participants' expertise affects the absolute prediction error between the prediction market and the average survey. In other words, does the advantage of prediction market relative to survey depend on the participants' expertise?

Reviewer: 3

Comments to the Author(s)

1. A main concern remains the relatively small sample size. A minimal target sample of 50 participants is too low and, consequently, the study should be regarded as very speculative. The language should be adjusted accordingly. Similarly, the statistical analysis should prominently address concerns due to a small number of independent observations while testing predictions about a relative large number of hypotheses (which now totalise to 22, with an increase of 57% relatively to the last version of this report).

2. Related to sample size, there is the issue of study power. In the current version, the estimated power has substantially increased with respect to the previous version of this report. However, it is unclear why this is the case, given that there is no substantial change in the observed correlations and the target sample size. Could it be an "artefact" of the increase in the number (from 14 to 22) of hypotheses? Or of the adoption of "Bayes factor"? Furthermore, the power analysis are made on the basis of prediction markets in the context of replications. This latter is arguably a different context from predicting new scientific results. Once again, the analysis should be taken as very speculative.

3. The introduction refers to Diego-Rosell et al. but it does not provide sufficient details to understand what they do and what the relationship is between their work and this study. The authors should rewrite (at least) the relevant paragraphs to make them sharper, more crystal-clear and self-contained.

4. It is unclear why the authors have made the shift from null hypothesis significance testing (p-value less than 0.05) to Bayes factor (of at least 10) to assess if a given hypothesis is supported. Arguably, Bayes factors are much less common in the statistical practices and, therefore, study participants may more likely be unfamiliar with them. Even more so, if we consider that the minimal requirement to be categorised as a researcher is to have a master degree or to be working on related topics. All of that again may be a factor hindering the robustness of the findings.

5. The incentives for participants appear quite bland. They are set in such a way that what matters is the average prediction made by participants. This is at odd with the hypothesis put forward in this report that focus on the individual predictions. It seems therefore that there is an issue of incentive compatibility.

6. The questions about researchers' academic background could be improved by making the categories better defined (for instance, considering possible differences across disciplines and/or the US system and beyond). Incidentally, the inclusion of students in dummy variable of the exploratory analyses seems to hint that the authors expect a significant number of researchers participating in the study to be actually student. If this would be the case, I guess it would be yet another instance of a factor hindering the robustness of the findings.

7. Responses to past commentaries were not provided and many past comments seemed to have been overlooked. Similarly, no rationale for the changes made was offered making the referee work more of a guessing game. The authors should take the time to address past comments and explicitly provide their response to reviewers' commentaries as well as how and – especially – why the manuscript has been revised.

Author's Response to Decision Letter for (RSOS-181308.R2)

See Appendix D.

Decision letter (RSOS-181308.R3)

23-Jan-2020

Dear Dr Dreber,

On behalf of the Editor, I am pleased to inform you that your amended Stage 1 Registered Report RSOS-181308.R3 (revised ad resubmitted after Stage 1 IPA) entitled "Using prediction markets to predict the outcomes in DARPA's Next Generation Social Science program" has been accepted in principle for publication in Royal Society Open Science.

We will ask for one additional minor revision, which you can make in the Stage 2 manuscript in due course. In the "Deviations from Protocol" section, please be sure to also explain the reason for each deviation and why it was necessary, rather than merely a description of the deviation as currently.

You may now progress to Stage 2 and complete the study as approved. Before commencing data collection we ask that you:

- 1) Update the journal office as to the anticipated completion date of your study.
- 2) Register your approved protocol on the Open Science Framework (<https://osf.io/>) or other recognised repository, either publicly or privately under embargo until submission of the Stage 2 manuscript. Please note that a time-stamped, independent registration of the protocol is mandatory under journal policy, and manuscripts that do not conform to this requirement cannot be considered at Stage 2. The protocol should be registered unchanged from its current approved state, with the time-stamp preceding implementation of the approved study design.

Following completion of your study, we invite you to resubmit your paper for peer review as a Stage 2 Registered Report. Please note that your manuscript can still be rejected for publication at Stage 2 if the Editors consider any of the following conditions to be met:

- The results were unable to test the authors' proposed hypotheses by failing to meet the approved outcome-neutral criteria.
- The authors altered the Introduction, rationale, or hypotheses, as approved in the Stage 1 submission.
- The authors failed to adhere closely to the registered experimental procedures. Please note that any deviations from the approved experimental procedures must be communicated to the editor

immediately for approval, and prior to the completion of data collection. Failure to do so can result in revocation of in-principle acceptance and rejection at Stage 2 (see complete guidelines for further information).

- Any post-hoc (unregistered) analyses were either unjustified, insufficiently caveated, or overly dominant in shaping the authors' conclusions.
- The authors' conclusions were not justified given the data obtained.

We encourage you to read the complete guidelines for authors concerning Stage 2 submissions at <https://royalsocietypublishing.org/rsos/registered-reports#ReviewerGuideRegRep>. Please especially note the requirements for data sharing, reporting the URL of the independently registered protocol, and that withdrawing your manuscript will result in publication of a Withdrawn Registration.

Once again, thank you for submitting your manuscript to Royal Society Open Science and we look forward to receiving your Stage 2 submission. If you have any questions at all, please do not hesitate to get in touch. We look forward to hearing from you shortly with the anticipated submission date for your stage two manuscript.

Kind regards,
Lianne Parkhouse
Editorial Coordinator
Royal Society Open Science
openscience@royalsociety.org

on behalf of Professor Chris Chambers (Registered Reports Editor, Royal Society Open Science)
openscience@royalsociety.org

Author's Response to Decision Letter for (RSOS-181308.R3)

See Appendix E.

RSOS-181308.R4 (Revision)

Review form: Reviewer 1

Is the manuscript scientifically sound in its present form?

Yes

Are the interpretations and conclusions justified by the results?

Yes

Is the language acceptable?

Yes

Do you have any ethical concerns with this paper?

No

Have you any concerns about statistical analyses in this paper?

No

Recommendation?

Accept with minor revision

Comments to the Author(s)

See attached file (Appendix F).

Review form: Reviewer 2

Is the manuscript scientifically sound in its present form?

Yes

Are the interpretations and conclusions justified by the results?

Yes

Is the language acceptable?

Yes

Do you have any ethical concerns with this paper?

No

Have you any concerns about statistical analyses in this paper?

No

Recommendation?

Accept as is

Comments to the Author(s)

I reviewed the research proposal version of this paper. The authors have addressed all my comments in the previous round. In the paper, the statistical analysis is sound, and the results are carefully discussed. Therefore, I would like to recommend acceptance.

Decision letter (RSOS-181308.R4)

Dear Dr Dreber:

On behalf of the Editor, I am pleased to inform you that your Stage 2 Registered Report RSOS-181308.R4 entitled "Using prediction markets to predict the outcomes in DARPA's Next Generation Social Science program" has been deemed suitable for publication in Royal Society Open Science subject to minor revision in accordance with the referee suggestions. Please find the referees' comments at the end of this email.

The reviewers and Subject Editor have recommended publication, but also suggest some minor revisions to your manuscript. Therefore, I invite you to respond to the comments and revise your manuscript.

Please also ensure that all the below editorial sections are included where appropriate -- if any section is not applicable to your manuscript, please can we ask you to nevertheless include the heading, but explicitly state that the heading is inapplicable. An example of these sections is attached with this email.

- Ethics statement

- Data accessibility

If you wish to submit your supporting data or code to Dryad (<http://datadryad.org/>), or modify your current submission to dryad, please use the following link:
[http://datadryad.org/submit?journalID=RSOS&manu=\(Document not available\)](http://datadryad.org/submit?journalID=RSOS&manu=(Document not available))

- Competing interests

- Authors' contributions

- Acknowledgements

- Funding statement

Because the schedule for publication is very tight, it is a condition of publication that you submit the revised version of your manuscript within 7 days (i.e. by the 19-Jun-2021). If you do not think you will be able to meet this date please let me know immediately.

Best regards,
Lianne Parkhouse
Editorial Coordinator

on behalf of Professor Chris Chambers
(Registered Reports Editor, Royal Society Open Science)
openscience@royalsociety.org

Associate Editor Comments to Author (Professor Chris Chambers):

Two of the original Stage 1 reviewers returned to assess the Stage 2 manuscript. As you will see, both are positive about the final paper, with Reviewer 2 recommending immediate acceptance and Reviewer 1 offering a number of very useful and constructive suggestions for optimising the presentation, as well as the clarity and usability of the online materials and data. Please attend carefully to these points in a revision and final Stage 2 acceptance should be forthcoming without requiring further in-depth review.

Reviewer Comments to Author:

Reviewer: 1

Comments to the Author(s)

See attached file

Reviewer: 2

Comments to the Author(s)

I reviewed the research proposal version of this paper. The authors have addressed all my comments in the previous round. In the paper, the statistical analysis is sound, and the results are carefully discussed. Therefore, I would like to recommend acceptance.

Author's Response to Decision Letter for (RSOS-181308.R4)

See Appendix G.

Decision letter (RSOS-181308.R5)

Dear Dr Dreber,

It is a pleasure to accept your Stage 2 Registered Report entitled "Using prediction markets to predict the outcomes in DARPA's Next Generation Social Science program" in its current form for publication in Royal Society Open Science.

on behalf of Professor Chris Chambers (Subject Editor)
openscience@royalsociety.org

Appendix A

Review of stage 1 registered report "Using prediction markets to address the incoherency problem in social science" by Anna Dreber et al.

2018-08-15

This review will be structured according to the headings given in the instruction to reviewers.

1. Significance of the research question(s)

The project aims to study whether prediction markets and surveys can predict which hypotheses will be supported in empirical social science experiments. This is an important question in light of reproducibility problems in science.

2. Logic, rationale, and plausibility of the proposed hypotheses

The proposed hypotheses draw on an ongoing research programme, in which the same team has conducted several prediction markets. The hypotheses are consistent with earlier research and theoretically plausible.

3. Soundness and feasibility of the methodology and analysis pipeline (including statistical power analysis where applicable)

The methodology of running a survey and prediction market is well-established, and has been shown by the authors to be feasible and effective in several earlier studies.

The power analysis gives a main estimate of 24%, which is so low that results will hardly be interpretable within a null hypothesis testing framework. As the authors note, the size of their proposed study is limited by the number of experiments ($n = 14$) included in the NGS2 project. They argue that even though statistical inferences from this study will not be very informative on their own, data may be pooled with other similar experiments to yield reliable inferences. I am persuaded by this argument. In my view, the authors are seizing a relatively rare opportunity to conduct a prediction market, adding to the body of existing data that already exists. Science does not need to progress in sliced blocks that each have a certain size.

This view however leads to two major implications which I suggest for consideration. First, I wonder if the authors have considered making the data available from all their prediction markets as a unified resource. This could be highly valuable, as opposed to sharing the data in different parcels for the different replication market experiments. Importantly, such an effort could increase reusability by providing harmonized variable names, and by identifying which participants are identical between datasets, in case someone wants to analyse individual differences in prediction market behaviour, for example.

Second, I wonder whether statistical inference tests such as those proposed can be justified, generally, in a low power context. If the main purpose of this study is to add data to a cumulative programme, might it not suffice to describe the data in detail and present descriptive statistics?

Assuming that the authors remain committed to inference tests, I would like to make the following remarks about the proposed statistical methods.

The analysis strategy rests primarily on correlation analyses. For instance, primary hypothesis 1 will be "[...] tested by a Spearman correlation, where we correlate prediction market prices with the binary outcome whether the hypotheses are supported or not." In this case, the independent unit of observation is each NGS2 experiment, each of which will have a prediction market price and a binary outcome. This is appropriate, as far as I can see. I would however suggest to rephrase the

hypotheses to reflect that the experiment and not the participant is the unit of observation, e.g. from "H0: Participants are not better than random in predicting whether the test results support the hypotheses or not with p-values less than 0.005 in the prediction markets." to "H0: Prediction market prices are not better than random in predicting whether test results support hypotheses or not with $p < 0.005$."

For primary hypotheses 3 and 4, each participant will rate each experiment (positive finding and effect size). While it is possible to average data over all participants within each experiment, this does not seem like the best strategy (nor can I find an explicit statement that this is what the authors plan to do). I suggest that the ratings be analysed with a mixed-effects model instead, as this will allow modelling of each participants' tendency to make ratings that are generally low or high.

It is not stated that any demographic or other information will be collected about the prediction market participants. It would be valuable to characterise the participants and record for instance how many of them are active researchers, how many work in the social sciences, what career stage they are at, etc.

A relatively minor point: under Participants, it is stated that in the event that NGS2 experiments are not completed, analyses will be adjusted. It should be clarified whether adjustment here means data exclusion, imputation, or something else.

The choice to use $\alpha < 0.005$ as a cutoff for statistical significance is consistent with the view expressed by several of the authors in a recent paper proposing to lower the threshold from 0.05 generally for new findings ("Redefine Statistical Significance"). While I am not altogether persuaded by the arguments presented in that paper, I do strongly believe that scientists are entitled to use any reasonable alpha level that they determine to be appropriate in a particular case.

4. Whether the clarity and degree of methodological detail would be sufficient to replicate exactly the proposed experimental procedures and analysis pipeline

The methodological detail is, strictly speaking, not sufficient in itself for exact replication, because the materials are not shared (I presume they may not have been written yet). The team has a good track record of sharing their materials from prediction market studies. I suggest to add a statement that the materials (i.e. the NGS2 hypotheses, survey questions etc.) be shared, possibly along with the data. Also, I would like to propose that the information to participants and consent forms be shared.

5. Whether the authors provide a sufficiently clear and detailed description of the methods to prevent undisclosed flexibility in the experimental procedures or analysis pipeline

Yes, excepting details noted above regarding the statistical analysis.

6. Whether the authors have considered sufficient outcome-neutral conditions (e.g. positive controls) for ensuring that the results obtained are able to test the stated hypotheses

In a study like this, the main thing to fear as far as I can see is that some participants could respond in an unconsidered manner, e.g. because they are tired of the survey and want to reach the end of it. The authors have decided not to exclude any data because those data appear to be suspect. I think they are right in doing so, because there is probably no good method to reliably decide whether the data are suspect or not.

Appendix B

Associate Editor Comments to Author (Professor Chris Chambers):

Associate Editor: 1

Comments to the Author:

Three expert reviewers have now assessed the submission. All find various degrees of merit in the proposal, while also offering a wide range of constructive suggestions for improvement. To highlight some of the key points: Reviewer 1 questions the necessarily low sample size and proposes a data aggregation strategy to increase the value of the research, together with the removal of all inferential statistical tests. From an editorial point of view, I am open to arguments either way in favour of significance testing in this context, but would invite the authors to clearly justify their approach. Reviewer 2 is positive but questions the level of financial incentive and, moreover, whether the design is sufficiently calibrated to maximise non-pecuniary incentives. This latter point is echoed by Reviewer 3 who is the most critical and, among wide range of concerns, also asks whether the research question is sufficiently substantive, whether the design accounts for order effects, and whether the sampling method is sufficiently informative (eg. whether further classification of participants is required). The reviewer also challenges the justification of the proposed hypotheses.

Overall I am convinced by the reviews that there is sufficient merit in the proposal to invite a substantial revision. Depending on the comprehensiveness of the response to reviewers, the manuscript may be returned to one or more of the reviewers for reassessment.

Dear Professor Chambers,

Many thanks for the reviews and for giving us the chance to revise our proposal. We have revised our proposal substantially and believe that we have now taken care of all the major concerns raised by the reviewers, and that this has led to a much improved proposal. Our comments to the reviewers will hereafter be in bold so that you can distinguish between the reviewers' comments and our comments.

Best wishes

Anna Dreber (on behalf of all coauthors)

Comments to Author:

Reviewer: 1

Review of stage 1 registered report "Using prediction markets to address the incoherency problem in social science" by Anna Dreber et al.
2018-08-15

This review will be structured according to the headings given in the instruction to reviewers.

1. Significance of the research question(s)

The project aims to study whether prediction markets and surveys can predict which hypotheses will be supported in empirical social science experiments. This is an important question in light of reproducibility problems in science.

Thank you!

2. Logic, rationale, and plausibility of the proposed hypotheses

The proposed hypotheses draw on an ongoing research programme, in which the same team has conducted several prediction markets. The hypotheses are consistent with earlier research and theoretically plausible.

Thanks for this positive assessment.

3. Soundness and feasibility of the methodology and analysis pipeline (including statistical power analysis where applicable)

The methodology of running a survey and prediction market is well-established, and has been shown by the authors to be feasible and effective in several earlier studies.

The power analysis gives a main estimate of 24%, which is so low that results will hardly be interpretable within a null hypothesis testing framework. As the authors note, the size of their proposed study is limited by the number of experiments ($n = 14$) included in the NGS2 project.

They argue that even though statistical inferences from this study will not be very informative on their own, data may be pooled with other similar experiments to yield reliable inferences. I am persuaded by this argument. In my view, the authors are seizing a relatively rare opportunity to conduct a prediction market, adding to the body of existing data that already exists. Science does not need to progress in sliced blocks that each have a certain size. This view however leads to two major implications which I suggest for consideration. First, I wonder if the authors have considered making the data available from all their prediction markets as a unified resource. This could be highly valuable, as opposed to sharing the data in different parcels for the different replication market experiments. Importantly, such an effort could increase reusability by providing harmonized variable names, and by identifying which participants are identical between datasets, in case someone wants to analyse individual differences in prediction market behaviour, for example.

This is a good idea. However, the data for all the projects are already available for anyone to use (on OSF), so we will think about how to do this in a separate project.

Second, I wonder whether statistical inference tests such as those proposed can be justified, generally, in a low power context. If the main purpose of this study is to add data to a cumulative programme, might it not suffice to describe the data in detail and present descriptive statistics?

We agree about the limited power of the hypotheses tests, but still prefer to keep them in the study. One could view reporting of these as presenting descriptive statistics (of the p-value of the tests, but interpreting results cautiously due to the limited number of observations).

Assuming that the authors remain committed to inference tests, I would like to make the following remarks about the proposed statistical methods.

The analysis strategy rests primarily on correlation analyses. For instance, primary hypothesis 1 will be "[...] tested by a Spearman correlation, where we correlate prediction market prices with the binary outcome whether the hypotheses are supported or not.". In this case, the independent unit of observation is each NGS2 experiment, each of which will have a prediction market price and a binary outcome. This is appropriate, as far as I can see. I would however suggest to rephrase the hypotheses to reflect that the experiment and not the participant is the unit of observation, e.g. from "H0: Participants are not better than random in predicting whether the test results support the hypotheses or not with p-values less than 0.005 in the prediction markets." to "H0: Prediction market prices are not better than random in predicting whether test results support hypotheses or not with $p < 0.005$."

Thanks for this suggestion of reframing the hypothesis, which we now have done.

For primary hypotheses 3 and 4, each participant will rate each experiment (positive finding and effect size). While it is possible to average data over all participants within each experiment, this does not seem like the best strategy (nor can I find an explicit statement that this is what the authors plan to do). I suggest that the ratings be analysed with a mixed-effects model instead, as this will allow modelling of each participants' tendency to make ratings that are generally low or high.

For primary hypotheses 3 and 4 we plan to average the data over all participants within each experiment, which has been clarified in the revised version. As the data for the prediction markets are by definition aggregated for each experiment (market), we prefer to analyze the survey data in the same aggregated manner so that it is straightforward to compare results between the prediction markets and the survey.

It is not stated that any demographic or other information will be collected about the prediction market participants. It would be valuable to characterise the participants and record for instance how many of them are active researchers, how many work in the social sciences, what career stage they are at, etc.

This is a great point. We will add a couple of questions on demographic information, but since we have no clear prior about how these variables correlate with predictions we will add this to the exploratory analysis.

A relatively minor point: under Participants, it is stated that in the event that NGS2 experiments are not completed, analyses will be adjusted. It should be clarified whether adjustment here means data exclusion, imputation, or something else.

This is a good point: We have now clarified this to mean that if an experiment is not completed, we will exclude this experiment from the analysis correlating prediction market beliefs and survey beliefs with actual outcomes from the experiments.

The choice to use $\alpha < 0.005$ as a cutoff for statistical significance is consistent with the view expressed by several of the authors in a recent paper proposing to lower the threshold from 0.05 generally for new findings ("Redefine Statistical Significance"). While I am not altogether persuaded by the arguments presented in that paper, I do strongly believe that scientists are entitled to use any reasonable alpha level that they determine to be appropriate in a particular case.

Thank you.

4. Whether the clarity and degree of methodological detail would be sufficient to replicate exactly the proposed experimental procedures and analysis pipeline
The methodological detail is, strictly speaking, not sufficient in itself for exact replication, because the materials are not shared (I presume they may not have been written yet). The team has a good track record of sharing their materials from prediction market studies. I suggest to add a statement that the materials (i.e. the NGS2 hypotheses, survey questions etc.) be shared, possibly along with the data. Also, I would like to propose that the information to participants and consent forms be shared.

This is a great point - we have added a sentence on this to the "Data accessibility" statement and will share all that information on the OSF page.

5. Whether the authors provide a sufficiently clear and detailed description of the methods to prevent undisclosed flexibility in the experimental procedures or analysis pipeline
Yes, excepting details noted above regarding the statistical analysis.

6. Whether the authors have considered sufficient outcome-neutral conditions (e.g. positive controls) for ensuring that the results obtained are able to test the stated hypotheses
In a study like this, the main thing to fear as far as I can see is that some participants could respond in an unconsidered manner, e.g. because they are tired of the survey and want to reach the end of it. The authors have decided not to exclude any data because those data appear to be suspect. I think they are right in doing so, because there is probably no good method to reliably decide whether the data are suspect or not.

We agree with the reviewer on this.

Reviewer: 2

Comments to the Author(s)

Review for "Using prediction markets to address the incoherency problem in social science"

Liangfei Qiu
University of Florida

This research proposes to examine a timely and important question: can prediction markets address the incoherency problem in social science? This research question has potential to contribute significantly to the prediction market literature and has important policy implications for academic research.

The proposed hypotheses are interesting and reasonable. In this proposal, the authors want to test if experts are better than random in predicting the results. I was wondering if another related hypothesis can also be tested: whether experts are better than normal people (e.g. college students) in predicting the results. In particular, the underlying mechanism that can effectively incentivize experts vs. normal people may be different.

We agree this would be interesting to test. Based on the recommendations of all three reviewers, we will now ask our participants for demographic information that will be analyzed in an exploratory analysis. We have added a question about background (academia versus non-academia) and a question about experience (the experience question is asked for each of the tested hypotheses), and we have added exploratory tests to see if these two characteristics are associated with the survey predictions (testing if the background and the experience are significantly associated with the absolute prediction error on survey predictions, in an individual data regression with clustering of standard errors on the individuals to take into account the correlation of multiple survey responses within individuals).

The methodology of the proposal is rigorous. The authors are very careful about interpreting their results. The description of the methods is clear and easy to follow. The authors provide sufficient details for readers to understand the research design. About research design, I was wondering if the monetary incentive (less than 100 dollars) is sufficient for the experts to provide thoughtful responses (these experts are not college students in typical laboratory experiments in economics and psychology)? Using non-pecuniary incentives may improve individual prediction accuracy in social-media based prediction markets. In other words, social effects and reputational concerns can play a key role in improving participants' prediction accuracy, and provide a powerful motivation for prediction market participants to supply thoughtful responses. If the participants (experts) are encouraged to share their predictions on social media, they may provide more thoughtful predictions.

In our previous prediction market studies participants have been anonymous, and showing the predictions on social media would imply losing anonymity. We have therefore refrained from adding this in this study. But this would be interesting to test in future studies and especially to compare such a “social media incentive” with a monetary incentive (but such a comparison would require a sizeable sample of results to predict).

Overall, I enjoyed reading this report and would like to recommend "Accept with minor revision."

Thank you!

Reviewer: 3

Comments to the Author(s)
Royal Society Open Science
Manuscript ID: RSOS-181308.

Title: Using prediction markets to address the incoherency problem in social science.

Authors: Anna Dreber, Yiling Chen, Magnus Johannesson, Brian Nosek, Thomas Pfeiffer and Adam Siegel.

Overall Assessment:

1. The registered report addresses an important research question, namely, “Could which research hypothesis receives empirical support or not be predicted ex-ante?” It is also quite topical given the surge of attention both inside and outside the scientific community about the ways in which the knowledge system works.

Thank you!

2. The authors repeatedly invite a judgement on the contribution of their proposed study in conjunction with the data they have collected in previous studies. While there is definitely value in investigating similar research questions to previous studies in a different domain, I would strongly invite the authors to enhance the innovative contribution of the proposed study by investigating other aspects, which were not into the focus of previous studies and/or may make the inter-study comparisons/integrations more informative.

This is a great point. One difference to previous studies, which we now emphasize more, is that this study does not predict replications outcomes. This study is about predicting results of new studies.

3. The hypotheses stated seem plausible though they could be made more sophisticated (see below). Similarly, their rationale could be detailed more.

Please see our responses below.

4. The methodology is overall sound. However, given the study design, I would be concerned ex-ante about possible order effects between the survey and the prediction markets that may affect the interpretation of any correlation between survey and market outcomes. I would recommend the author to make appropriate design changes to control for order effects.

This is another very good point. However, the difficulty with changing or varying the order between the survey and the prediction market is that participating in the

prediction markets gives participants information about the beliefs of other researchers about the tested hypotheses (to share and disseminate such information across traders is one mechanism for the markets to provide accurate predictions). One would therefore expect that participating in the prediction markets should affect survey answers; so this would not be a pure order effect but would measure to what extent researchers update information from participating in the markets. Filling out the survey on the other hand reveals no information about the beliefs of other researchers. To test if filling out the survey still has an effect on market predictions, one would need one treatment with a survey followed by prediction markets and one treatment with only prediction markets. However, that would require twice as many participants and is beyond the scope of the present study but this would be great to explore in future studies with larger sample sizes. We will acknowledge this issue in the limitations discussion of the final paper.

5. In general, the details of (and sometimes the motivations for) the study methodology are lacking in this report (though they could be inferred from some of the studies referenced). Incidentally, it would have been useful to include supplemental material to provide more details about the study methodology.

We agree and have added more information about this in the main text of the revised version.

6. Hopefully, the authors will find my commentary useful and may be able to improve further their proposed study and deliver interesting findings for the scientific community.

Thank you for your suggestions!

Major Comments:

7. Both the title and the abstract do not seem to reflect fully the main body of the report (e.g., reference to the "incoherence problem").

We agree and have revised the title and the Abstract. The title is now "Using prediction markets to predict the outcomes in DARPA's Next Generation Social Science program". The abstract has also been updated.

8. The introduction – rightly so – puts at centre stage the DARPA's Next Generation Social Science (NGS2) programme. However, I fear that for readers without prior familiarity with the programme the information provided by the authors is too limited and somewhat airy to understand fully the context of the proposed study. I feel the authors should make an additional effort to communicate the core elements of the programme and provide the reader with more details either in a footnote or in an appendix (or both).

We agree and have provided more information about this. We now discuss the relevant NGS2 study (Diego-Rosell et al.) for which we are eliciting peer beliefs.

9. The final paragraph of the introduction defines a hypothesis as 'supported' by relying on a p-value less than 0.005. It is unclear why this should be the relevant threshold. Why not the canonical 0.05? I think the authors should motivate their choice of a threshold in detail by also arguing as much as possible how such a choice will not impair the inference that can be drawn by the study.

We have motivated this further in the revised version, mainly from discussing the recent paper by Benjamin et al. (2018). Mainly this is about communicating our results

of the study in a responsible manner and not overinterpret our findings. Note also that we use both the 0.005 and the 0.05 thresholds for interpreting our results, but refer to results below the 0.05 threshold as “suggestive evidence” and results below the more stringent 0.005 threshold as “statistically significant evidence”.

10. The authors intend to recruit participants via Brian Nosek’s twitter feed. This recruitment strategy may allow reaching a wider audience, but it does not allow precise participant targeting. (I guess non-researchers may also take part in the study). Therefore, while it is fine not to exclude any participant/discard any data, it would be insightful to learn about participants’ types. Indeed, ex-ante I would expect different predictions coming from either researchers or lay people; and, among researchers, different predictions coming from those closer to the field of investigation of the hypotheses being tested relatively to those coming from researchers further away. I would invite the authors to elicit background characteristics, which may allow them to classify participants in types relevant to the prediction exercise.

We also think that demographic information may be interesting and based on the comments from all three reviewers we have now chosen to collect this and add this to the exploratory analysis (please see our reply to Reviewer 2 for more).

11. The lottery incentive system for the survey measures seems – in expected terms – quite bland. Moreover, some assessment of belief elicitation methods points towards the direction of mixed effects in using incentives when eliciting beliefs (e.g., Schotter and Trevino, 2014). Why not running the survey without monetary incentives? Would not be best to not pay at all (perhaps, just keeping a flat fee for participation) if paying enough is not feasible (e.g., Gneezy and Rustichini, 2000)? After all, the prediction market should be the mechanism where monetary incentives are at play to elicit beliefs in an incentive-compatible way.

We have followed this recommendation and removed the incentives from the survey.

12. The prediction markets will be open for two weeks. However, it could happen that in such a time window only few people will volunteer to take part in the study. The authors should commit to a minimum sample size to consider the study as successful implemented for leading to a Royal Society Open Science publication. They should also argue why such a sample size is the right as minimum sample size and what alternative plans they will put in place in case the minimum sample size is not met within the two-week window.

We agree and have added the requirement that at least 50 subjects participate in the prediction market part; this is somewhat higher than in the Dreber et al. (PNAS 2015) prediction markets on RPP replications.

13. Not all participants may easily understand how prediction market and its web-based interface work. I would recommend the authors to build in opportunities for the participants to familiarise with the functioning of the market and its interface as well as ways of testing their understanding. Any successful attempt in this direction may reduce noise in the data and enhance the robustness of the interpretation of the empirical evidence.

We use extensive instructions on the functioning of the markets based on our previous studies that seem to work well, and participants also learn by actively participating in the markets.

14. It seems that the benchmark for hypothesis testing of a random prediction is a too easy target and not a very informative one. I would urge the authors to elaborate further (for instance on a 'scientifically significant' difference relative to a random prediction) and formulate more challenging and informative benchmarks.

Given the relatively low number of markets (n=14), and the limited statistical power in detecting a difference to random predictions, we have refrained from formulating a more demanding hypothesis (as it would be underpowered).

15. Integrate the classical hypothesis testing with alternative approaches and/or econometric analysis would also constitute added value.

We agree, but with our target sample size (at least 50 participants in the prediction markets) the statistical power to detect effects in additional econometric analysis is limited. We have added one additional econometric analysis: testing testing if the absolute prediction error on the survey predictions is significantly associated with background and experience (see our response to comment 10 above).

Minor Comments:

16. Page 3, row 20: the acronym OSF appears without being introduced.

We have corrected this.

17. Page 4, row 23: I would invite the authors to use a different letter for the binary variable rather than p^* : indeed, p is a letter usually used to indicate probabilities and employing it for the binary variable may generate unnecessary, though frankly limited, confusion.

Good point - since we have deleted this part though we did not do anything.

18. Page 4, row 51: there is a typo "maker" should read "market".

Thanks, we have corrected this.

References

Gneezy, Uri, and Aldo Rustichini. 2000. Pay Enough or Don't Pay at All. *The Quarterly Journal of Economics*, 115 (3), 791–810.

Schotter, Andrew, and Isabel Trevino. 2014. Belief Elicitation in the Laboratory. *Annual Review of Economics*, 6(1), 103-128.

Appendix C

Dear Editorial Team and Professor Chambers,

We have had some unplanned changes to our project (Manuscript ID RSOS-181308.R2). Originally, our plan was to run prediction markets and surveys trying to forecast results from a study that was part of DARPA's Next Generation Social Science program Cycle 2. After having received "accepted in principle for publication" in RSOS, we were getting ready to launch the markets and surveys in the spring of 2019. But it turned out the results from those experiments were already public, so we could not go ahead as planned.

But we instead got the opportunity to predict outcomes for the next cycle of the DARPA's Next Generation Social Science program (Cycle 3). Our hypotheses that we are testing are exactly the same as before, but the differences are the following:

- 1) We are predicting experimental results from Cycle 3 instead of Cycle 2.
- 2) Instead of predicting binary outcomes in terms of whether a result is statistically significant ($p < 0.05$) or not, we predict whether the Bayes Factor is at least 10 or not (with details around this, from the pre-analysis plan of the Cycle 3 project).
- 3) We now have incentivized questions in the survey (instead of no incentives).
- 4) Instead of having 11 hypotheses, we have 22 hypotheses, thus our statistical power is higher.
- 5) We have added two of the NGS2 Cycle 3 researcher to the project since collaborating with them has been key in figuring out what hypotheses they are testing. We have also added a postdoc (Domenico Viganola) to the project.
- 6) We have added a robustness test to the Secondary Hypotheses.

We are submitting a marked up version of the new manuscript to indicate how it deviates from the accepted in principle version. We have recently performed the surveys and the prediction markets soon since the Cycle 3 researchers will need to release their results fairly soon. The original pre-analysis plan was posted on OSF 2019/10/16 (before the launch of the survey and the prediction markets and thus before the data collection) and it has been amended on 2019/11/05 (before the markets had closed): We noticed a typo in the formulation of hypotheses 1 and 3: the definition of 'hypothesis is supported' was not consistent with the rest of the document, so we deleted "with p-values less than 0.005" since these hypotheses are about Bayes Factors. We amended the pre-analysis plan on 2019/11/05 before the conclusion of the data collection phase. This is the registered report that we here are submitting.

Many thanks for your input, and sorry about this (but nice that we have higher power than planned!).

Best wishes

Anna Dreber Almenberg

Appendix D

Dear Professor Chambers,

Many thanks for giving us the opportunity to revise our manuscript. The suggestions from you and the reviewers are excellent and we believe that the manuscript has improved. Please see our comments to the reviewers below marked in bold, as well as the revised manuscript.

Best wishes

Anna Dreber (on behalf of all coauthors)

Associate Editor Comments to Author (Professor Chris Chambers):

This is an unusual situation involving the reassessment of an amended Stage 1 protocol following IPA. The three original reviewers who assessed the original Stage 1 manuscript have now completed their reassessments. Reviewers 1 and 2 are broadly positive, although Reviewer 1 recommends rejection on essentially administrative grounds. The reviewer makes an important point about the the amended protocol potentially conflicting with the existing preregistration (which is an issue that I can see confusing readers), but rather than issuing a procedural rejection, I would like to offer the authors the opportunity to anticipate and respond to this concern in the manuscript using a 'Deviations from Protocol' section, ensuring that the changes are comprehensively explained and justified. Reviewer 2 is similarly positive but offers some minor recommendations for revision, including stronger justification of the recruitment mode. Reviewer 3 remains negatively inclined, and although I was sufficiently convinced by the authors' previous responses to offer IPA, the reactivation of the Stage 1 review process provides the opportunity to address these concerns (and the reviewer's additional reasonable concerns) more adequately than in the previous version of the manuscript. Based on this combination of reviews, a Major Revision is recommended.

We understand that the amended protocol could confuse readers. We think the idea to have a 'Deviations from Protocol' section is a great solution so we have now included this.

Reviewer Comments to Author:

Reviewer: 1

Comments to the Author(s)

Thank you for the opportunity to re-review this manuscript. After stage 1 review and in-principle acceptance, the study has changed substantially. The authors provide clear and compelling explanations and motivations for the changes. However, a preregistration was made on OSF, and data collection is now completed under that preregistration. This means that any further requests for revisions at stage 1 will not be able to influence data collection, and may conflict with the authors' preregistration. Therefore, I question whether this manuscript should be re-reviewed at stage 1, rather than either moving to stage 2 or withdrawing the registration. I remain convinced that this work makes a valuable contribution, and I look forward to the editor's appraisal. Since a recommendation is required, I have made a pro forma entry of "reject", based on the procedural concerns outlined above and not on the substantive content of the manuscript.

Reviewer: 2

Comments to the Author(s)

I have reviewed the previous version of the paper. It studies an interesting and timely research question. The paper is written well, and the research method is rigorous. I have the following comments, and I hope they are helpful.

1. Participants will be recruited through social media via Brian Nosek's twitter feed and other behavioral science organizations. Why the participants will be recruited using a particular twitter feed (instead of sending emails to a given set of researchers)? The authors may want to justify it.

This is a good point - given Brian Nosek's large following on Twitter of researchers who are interested in reproducibility we thought this would be a good way to recruit. We have added this information.

2. Potentially, the authors could investigate how demographic information affects the hypotheses. In particular, I am interested in how the participants' expertise affects the absolute prediction error between the prediction market and the average survey. In other words, does the advantage of prediction market relative to survey depend on the participants' expertise?

It is straightforward to test if the two demographic variables (academia vs non-academia and expertise in the hypotheses) are associated with the individual absolute prediction error on survey predictions, and we have this in the Exploratory analyses. Doing this for the prediction market unfortunately cannot be done as the market data is on the aggregate and not the individual level (so we cannot estimate a prediction error on the individual level for the markets).

Reviewer: 3

Comments to the Author(s)

1. A main concern remains the relatively small sample size. A minimal target sample of 50 participants is too low and, consequently, the study should be regarded as very speculative. The language should be adjusted accordingly. Similarly, the statistical analysis should prominently address concerns due to a small number of independent observations while testing predictions about a relative large number of hypotheses (which now totalise to 22, with an increase of 57% relatively to the last version of this report).

We agree that a larger sample would be preferable everything else equal. However, we use this minimal target sample size largely based on our previous work. In this previous work, we have had researchers predict whether studies replicate where a successful replication is defined as an effect in the same direction as the original paper with a p-value less than 0.05. In this previous work we have used both surveys and prediction markets. In the survey we ask participants to give a probability to whether the study replicates or not (thus binary outcome). In the prediction markets, we let participants trade contracts worth \$1 if the study replicates and \$0 if it does not replicate. Prices start at 50 cents, thus similar to the current setup. For 44 studies in the RPP (out of which 41 studies were completed), we had two sets of surveys and prediction markets (Dreber et al. 2015). In the first set for 23 replication studies we had 47 active participants, and in the second set for 21 replication studies we had 45 active participants. The number of active traders per market (where one market is a replication) ranged from 18 to 40. We also had a survey and prediction markets for the 18 studies replicated in the Experimental Economics Replication Project (Camerer et al. 2016), with a sample size of 97 participants; the 21 studies in the “Evaluating the replicability of social science experiments in Nature and Science between 2010 and 2015” (Camerer et al. 2018). Here we had two conditions, with 114 and 92 participants in each treatment; the 24 studies in the “Predicting replication outcomes in the Many Labs 2 study” (Forsell et al.), with a sample size of 78 participants.

It is also interesting to note that DellaVigna and Pope (in press) note that “a couple dozen respondents are enough to achieve the wisdom-of-the-crowd effect” (pg. 15) in their forecasting study.

2. Related to sample size, there is the issue of study power. In the current version, the estimated power has substantially increased with respect to the previous version of this report. However, it is unclear why this is the case, given that there is no substantial change in the observed correlations and the target sample size. Could it be an “artefact” of the increase in the number (from 14 to 22) of hypotheses? Or of the adoption of “Bayes factor”? Furthermore, the power analysis are made on the basis of prediction markets in the context of replications. This latter is arguably a different context from predicting new scientific results. Once again, the analysis should be taken as very speculative.

We have done the power analysis based on the number of NGS2 studies included - thus power has increased because of the increase in the number of studies. We now make it clearer that the power analysis is based on previous results mainly in the context of

replications but also in the context of new scientific results, while we in the current project are predicting new scientific results.

3. The introduction refers to Diego-Rosell et al. but it does not provide sufficient details to understand what they do and what the relationship is between their work and this study. The authors should rewrite (at least) the relevant paragraphs to make them sharper, more crystal-clear and self-contained.

This is a very good point and we have updated this.

4. It is unclear why the authors have made the shift from null hypothesis significance testing (p -value less than 0.05) to Bayes factor (of at least 10) to assess if a given hypothesis is supported. Arguably, Bayes factors are much less common in the statistical practices and, therefore, study participants may more likely be unfamiliar with them. Even more so, if we consider that the minimal requirement to be categorised as a researcher is to have a master degree or to be working on related topics. All of that again may be a factor hindering the robustness of the findings.

In our work, we still stick to null hypothesis significance testing. However, Diego-Rosell et al. shifted to Bayes factors since they thought it made more sense for their particular project. Since Bayes factors are less common we emphasize in the recruitment material that participants will make predictions on Bayes factors. We have made this clear in the manuscript.

5. The incentives for participants appear quite bland. They are set in such a way that what matters is the average prediction made by participants. This is at odds with the hypothesis put forward in this report that focus on the individual predictions. It seems therefore that there is an issue of incentive compatibility.

In terms of general incentives in the prediction markets - the prediction markets are incentive compatible and the size of the incentives are similar to what we have used in the past thus we have strong reasons to believe that they are sufficient. In the survey we have in the past typically not used any monetary incentives, thus we believe that adding some incentives might if anything improve the survey responses. To incentivize the effect size forecasts in the surveys we do not use the average prediction, but the average over the squared forecasting errors.

6. The questions about researchers' academic background could be improved by making the categories better defined (for instance, considering possible differences across disciplines and/or the US system and beyond). Incidentally, the inclusion of students in dummy variable of the exploratory analyses seems to hint that the authors expect a significant number of researchers participating in the study to be actually student. If this would be the case, I guess it would be yet another instance of a factor hindering the robustness of the findings.

Since we have already collected the data, we unfortunately cannot change these questions. We think most participants will not be students, and that most students will be PhD students, but this remains to be seen.

7. Responses to past commentaries were not provided and many past comments seemed to have been overlooked. Similarly, no rationale for the changes made was offered making the referee work more of a guessing game. The authors should take the time to address past comments and explicitly provide their response to reviewers' commentaries as well as how and – especially – why the manuscript has been revised.

We have provided responses to previously received comments. We apologize that it was not clear why the main changes were made to the manuscript - these were due to changes in the experiments we set out to predict.

Appendix E

Dear Professor Chambers,

Many thanks for giving us the opportunity to revise our manuscript. The suggestions from you and the reviewers are excellent and we believe that the manuscript has improved. Please see our comments to the reviewers below marked in bold, as well as the revised manuscript.

Best wishes

Anna Dreber (on behalf of all coauthors)

Associate Editor Comments to Author (Professor Chris Chambers):

This is an unusual situation involving the reassessment of an amended Stage 1 protocol following IPA. The three original reviewers who assessed the original Stage 1 manuscript have now completed their reassessments. Reviewers 1 and 2 are broadly positive, although Reviewer 1 recommends rejection on essentially administrative grounds. The reviewer makes an important point about the the amended protocol potentially conflicting with the existing preregistration (which is an issue that I can see confusing readers), but rather than issuing a procedural rejection, I would like to offer the authors the opportunity to anticipate and respond to this concern in the manuscript using a 'Deviations from Protocol' section, ensuring that the changes are comprehensively explained and justified. Reviewer 2 is similarly positive but offers some minor recommendations for revision, including stronger justification of the recruitment mode. Reviewer 3 remains negatively inclined, and although I was sufficiently convinced by the authors' previous responses to offer IPA, the reactivation of the Stage 1 review process provides the opportunity to address these concerns (and the reviewer's additional reasonable concerns) more adequately than in the previous version of the manuscript. Based on this combination of reviews, a Major Revision is recommended.

We understand that the amended protocol could confuse readers. We think the idea to have a 'Deviations from Protocol' section is a great solution so we have now included this.

Reviewer Comments to Author:

Reviewer: 1

Comments to the Author(s)

Thank you for the opportunity to re-review this manuscript. After stage 1 review and in-principle acceptance, the study has changed substantially. The authors provide clear and compelling explanations and motivations for the changes. However, a preregistration was made on OSF, and data collection is now completed under that preregistration. This means that any further requests for revisions at stage 1 will not be able to influence data collection, and may conflict with the authors' preregistration. Therefore, I question whether this manuscript should be re-reviewed at stage 1, rather than either moving to stage 2 or withdrawing the registration. I remain convinced that this work makes a valuable contribution, and I look forward to the editor's appraisal. Since a recommendation is required, I have made a pro forma entry of "reject", based on the procedural concerns outlined above and not on the substantive content of the manuscript.

Reviewer: 2

Comments to the Author(s)

I have reviewed the previous version of the paper. It studies an interesting and timely research question. The paper is written well, and the research method is rigorous. I have the following comments, and I hope they are helpful.

1. Participants will be recruited through social media via Brian Nosek's twitter feed and other behavioral science organizations. Why the participants will be recruited using a particular twitter feed (instead of sending emails to a given set of researchers)? The authors may want to justify it.

This is a good point - given Brian Nosek's large following on Twitter of researchers who are interested in reproducibility we thought this would be a good way to recruit. We have added this information.

2. Potentially, the authors could investigate how demographic information affects the hypotheses. In particular, I am interested in how the participants' expertise affects the absolute prediction error between the prediction market and the average survey. In other words, does the advantage of prediction market relative to survey depend on the participants' expertise?

It is straightforward to test if the two demographic variables (academia vs non-academia and expertise in the hypotheses) are associated with the individual absolute prediction error on survey predictions, and we have this in the Exploratory analyses. Doing this for the prediction market unfortunately cannot be done as the market data is on the aggregate and not the individual level (so we cannot estimate a prediction error on the individual level for the markets).

Reviewer: 3

Comments to the Author(s)

1. A main concern remains the relatively small sample size. A minimal target sample of 50 participants is too low and, consequently, the study should be regarded as very speculative. The language should be adjusted accordingly. Similarly, the statistical analysis should prominently address concerns due to a small number of independent observations while testing predictions about a relative large number of hypotheses (which now totalise to 22, with an increase of 57% relatively to the last version of this report).

We agree that a larger sample would be preferable everything else equal. However, we use this minimal target sample size largely based on our previous work. In this previous work, we have had researchers predict whether studies replicate where a successful replication is defined as an effect in the same direction as the original paper with a p-value less than 0.05. In this previous work we have used both surveys and prediction markets. In the survey we ask participants to give a probability to whether the study replicates or not (thus binary outcome). In the prediction markets, we let participants trade contracts worth \$1 if the study replicates and \$0 if it does not replicate. Prices start at 50 cents, thus similar to the current setup. For 44 studies in the RPP (out of which 41 studies were completed), we had two sets of surveys and prediction markets (Dreber et al. 2015). In the first set for 23 replication studies we had 47 active participants, and in the second set for 21 replication studies we had 45 active participants. The number of active traders per market (where one market is a replication) ranged from 18 to 40. We also had a survey and prediction markets for the 18 studies replicated in the Experimental Economics Replication Project (Camerer et al. 2016), with a sample size of 97 participants; the 21 studies in the “Evaluating the replicability of social science experiments in Nature and Science between 2010 and 2015” (Camerer et al. 2018). Here we had two conditions, with 114 and 92 participants in each treatment; the 24 studies in the “Predicting replication outcomes in the Many Labs 2 study” (Forsell et al.), with a sample size of 78 participants.

It is also interesting to note that DellaVigna and Pope (in press) note that “a couple dozen respondents are enough to achieve the wisdom-of-the-crowd effect” (pg. 15) in their forecasting study.

2. Related to sample size, there is the issue of study power. In the current version, the estimated power has substantially increased with respect to the previous version of this report. However, it is unclear why this is the case, given that there is no substantial change in the observed correlations and the target sample size. Could it be an “artefact” of the increase in the number (from 14 to 22) of hypotheses? Or of the adoption of “Bayes factor”? Furthermore, the power analysis are made on the basis of prediction markets in the context of replications. This latter is arguably a different context from predicting new scientific results. Once again, the analysis should be taken as very speculative.

We have done the power analysis based on the number of NGS2 studies included - thus power has increased because of the increase in the number of studies. We now make it clearer that the power analysis is based on previous results mainly in the context of

replications but also in the context of new scientific results, while we in the current project are predicting new scientific results.

3. The introduction refers to Diego-Rosell et al. but it does not provide sufficient details to understand what they do and what the relationship is between their work and this study. The authors should rewrite (at least) the relevant paragraphs to make them sharper, more crystal-clear and self-contained.

This is a very good point and we have updated this.

4. It is unclear why the authors have made the shift from null hypothesis significance testing (p-value less than 0.05) to Bayes factor (of at least 10) to assess if a given hypothesis is supported. Arguably, Bayes factors are much less common in the statistical practices and, therefore, study participants may more likely be unfamiliar with them. Even more so, if we consider that the minimal requirement to be categorised as a researcher is to have a master degree or to be working on related topics. All of that again may be a factor hindering the robustness of the findings.

In our work, we still stick to null hypothesis significance testing. However, Diego-Rosell et al. shifted to Bayes factors since they thought it made more sense for their particular project. Since Bayes factors are less common we emphasize in the recruitment material that participants will make predictions on Bayes factors. We have made this clear in the manuscript.

5. The incentives for participants appear quite bland. They are set in such a way that what matters is the average prediction made by participants. This is at odds with the hypothesis put forward in this report that focus on the individual predictions. It seems therefore that there is an issue of incentive compatibility.

In terms of general incentives in the prediction markets - the prediction markets are incentive compatible and the size of the incentives are similar to what we have used in the past thus we have strong reasons to believe that they are sufficient. In the survey we have in the past typically not used any monetary incentives, thus we believe that adding some incentives might if anything improve the survey responses. To incentivize the effect size forecasts in the surveys we do not use the average prediction, but the average over the squared forecasting errors.

6. The questions about researchers' academic background could be improved by making the categories better defined (for instance, considering possible differences across disciplines and/or the US system and beyond). Incidentally, the inclusion of students in dummy variable of the exploratory analyses seems to hint that the authors expect a significant number of researchers participating in the study to be actually student. If this would be the case, I guess it would be yet another instance of a factor hindering the robustness of the findings.

Since we have already collected the data, we unfortunately cannot change these questions. We think most participants will not be students, and that most students will be PhD students, but this remains to be seen.

7. Responses to past commentaries were not provided and many past comments seemed to have been overlooked. Similarly, no rationale for the changes made was offered making the referee work more of a guessing game. The authors should take the time to address past comments and explicitly provide their response to reviewers' commentaries as well as how and – especially – why the manuscript has been revised.

We have provided responses to previously received comments. We apologize that it was not clear why the main changes were made to the manuscript - these were due to changes in the experiments we set out to predict.

Appendix F

Review of "Using prediction markets to predict the outcomes in DARPA's Next Generation Social Science program", stage 2, by Viganola et al, for RSOS

Thank you for the opportunity to review this stage 2 registered report. I have previously reviewed this report at stage 1.

Consistency of stage 2 report with stage 1 report

The introduction, rationale and stated hypotheses are the same as in the approved Stage 1 submission. The authors have adhered to the registered experimental procedures. Deviations are clearly reported, well justified, and informative. The increase in sample size is a particular improvement. I recommend that the relationship of the study to the existing preregistration on OSF should be declared and explained as well.

Reporting of results

The results are overall reported clearly and in accordance with the stage 1 report. Additional analyses not planned in stage 1 are clearly identified as such.

The authors have chosen an alpha level of 0.005 throughout, but this is not consistently applied. In some places statistical significance is claimed for p-values in the interval 0.05—0.005, and in one place a p-value is reported as equal to 0.005 with no indication whether it was rounded up or down.

Figure 1 is clear and helpful in conveying the results. I suggest the authors add plots for effect sizes as well (possibly a set of 3 plots). As a very minor detail, I suggest that since the same variable is plotted on the x and y axes, it could be helpful to have an aspect ratio of 1 (i.e. to make the plotting area a square), and to add a diagonal line representing $x = y$, against which the observed difference can be compared.

The discussion section is brief and makes no claims that are unsupported by the data. This is fine with me. I am left with a curiosity about what the authors think we have learned, and whether in their opinion the outstanding research agenda in this field has moved/changed due to the observed results.

Materials, data, and code

I have reviewed the OSF project page on which study materials including data and code have been posted. I particularly appreciated that the IRB application and approval are posted there.

Due in part to the limited sample size, the authors envisaged at the outset that the data from this project would be analysed in the future together with other datasets from similar studies. Partly for that reason, the data are in my view maybe the most important output of this project. Therefore, I recommend that care be taken to ensure that the data are managed and shared well. Currently, the data meet several best practices: data are mainly available in csv files, and fairly well structured, the folder contains a readme which describes the files etc. But there is scope for improvement. I have the following recommendations:

- The readme file should describe every file in the folder (or the folder should be cleaned up). There is currently an undeclared working draft of the manuscript in there.
- Descriptions in the readme file should be clear and understandable to an outsider.
- Data files should have a codebook/data dictionary that lists the variables and explains what they mean.
- It is apparent from the data files that results from NGS2 were re-evaluated at some point and some were determined to be inconclusive (rather than supported or unsupported). The

manuscript makes no mention of this, which leaves the reader wondering if it could be a limitation. An explanation would help.

- The whole project should be archived on OSF by creating a registration, in order to ensure all the components will remain accessible in the future.

The analysis code in R is well structured and reasonably well commented. I was able to follow the structure of how data wrangling and analyses were conducted. While I did not attempt to re-run the code on my computer, I did not identify any obvious problems or red flags. I recommend to state in the manuscript which versions were used of R and the package "miceadds", and that these be cited in the reference list.

Appendix G

Dear Editors and Coordinators,

Many thanks for giving us the opportunity to revise the manuscript. We have responded to all the comments – please see below for more information - and we also include all editorial sections where appropriate.

Best wishes

Anna (on behalf of all authors)

Review of "Using prediction markets to predict the outcomes in DARPA's Next Generation Social Science program", stage 2, by Viganola et al, for RSOS

Thank you for the opportunity to review this stage 2 registered report. I have previously reviewed this report at stage 1.

Consistency of stage 2 report with stage 1 report

The introduction, rationale and stated hypotheses are the same as in the approved Stage 1 submission. The authors have adhered to the registered experimental procedures. Deviations are clearly reported, well justified, and informative. The increase in sample size is a particular improvement. I recommend that the relationship of the study to the existing preregistration on OSF should be declared and explained as well.

Our reply: We have now clarified this further in the "Deviations from Protocol" paragraph.

Reporting of results

The results are overall reported clearly and in accordance with the stage 1 report. Additional analyses not planned in stage 1 are clearly identified as such. The authors have chosen an alpha level of 0.005 throughout, but this is not consistently applied. In some places statistical significance is claimed for p-values in the interval 0.05—0.005, and in one place a p-value is reported as equal to 0.005 with no indication whether it was rounded up or down.

Our reply: Thanks for catching that error – we are now consistent with the term statistical significance and also make it clear that the 0.005 value is rounded from 0.0053 (by calling it suggestive evidence).

Figure 1 is clear and helpful in conveying the results. I suggest the authors add plots for effect sizes as well (possibly a set of 3 plots). As a very minor detail, I suggest that since the same variable is plotted on the x and y axes, it could be helpful to have an aspect ratio of 1 (i.e. to make the plotting area a square), and to add a diagonal line representing $x = y$, against which the observed difference can be compared.

Our reply: This is a great idea and we have updated the figure so that it now contains 3 plots.

The discussion section is brief and makes no claims that are unsupported by the data. This is fine with me. I am left with a curiosity about what the authors think we have learned, and whether in their opinion the outstanding research agenda in this field has moved/changed due to the observed results.

Our reply: We have added a sentence on this.

Materials, data, and code

I have reviewed the OSF project page on which study materials including data and code have been posted. I particularly appreciated that the IRB application and approval are posted there. Due in part to the limited sample size, the authors envisaged at the outset that the data from this project would be analysed in the future together with other datasets from similar studies. Partly for that reason, the data are in my view maybe the most important output of this project. Therefore, I recommend that care be taken to ensure that the data are managed and shared well. Currently, the data meet several best practices: data are mainly available in csv files, and fairly well structured, the folder contains a readme which describes the files etc. But there is scope for improvement. I have the following recommendations:

- The readme file should describe every file in the folder (or the folder should be cleaned up). There is currently an undeclared working draft of the manuscript in there.
- Descriptions in the readme file should be clear and understandable to an outsider.
- Data files should have a codebook/data dictionary that lists the variables and explains what they mean.
- It is apparent from the data files that results from NGS2 were re-evaluated at some point and some were determined to be inconclusive (rather than supported or unsupported). The manuscript makes no mention of this, which leaves the reader wondering if it could be a limitation. An explanation would help.
- The whole project should be archived on OSF by creating a registration, in order to ensure all the components will remain accessible in the future.

The analysis code in R is well structured and reasonably well commented. I was able to follow the structure of how data wrangling and analyses were conducted. While I did not attempt to re-run the code on my computer, I did not identify any obvious problems or red flags. I recommend to state in the manuscript which versions were used of R and the package "miceadds", and that these be cited in the reference list.

Our reply: These are excellent points. We have updated the folder with all codes and clarifications accordingly, and we also cite the R version used and the package "miceadds".